# CD276-dependent efferocytosis by tumor-associated macrophages promotes immune evasion in bladder cancer

Maosheng Cheng[1,8], Shuang Chen[1,8], Kang Li[1,8], Ganping Wang[2,8], Gan Xiong[1,8], Rongsong Ling [3], Caihua Zhang[1], Zhihui Zhang[1], Hui Han [1], Zhi Chen[1], Xiaochen Wang[1], Yu Liang[1], Guoli Tian[4], Ruoxing Zhou[1], Yan Zhu[1], Jieyi Ma[1], Jiahong Liu[5], Shuibin Lin [1] ✉, Hao Xu [6] ✉, Demeng Chen [1] ✉, Yang Li [7] ✉ & Liang Peng[5] ✉

Interplay between innate and adaptive immune cells is important for the antitumor immune response. However, the tumor microenvironment may turn immune suppressive, and tumor associated macrophages are playing a role in this transition. Here, we show that CD276, expressed on tumor-associated macrophages (TAM), play a role in diminishing the immune response against tumors. Using a model of tumors induced by N-butyl-N-(4-hydroxybutyl) nitrosamine in BLCA male mice we show that genetic ablation of CD276 in TAMs blocks efferocytosis and enhances the expression of the major histocompatibility complex class II (MHCII) of TAMs. This in turn increases CD4 + and cytotoxic CD8 + T cell infiltration of the tumor. Combined single cell RNA sequencing and functional experiments reveal that CD276 activates the lysosomal signaling pathway and the transcription factor JUN to regulate the expression of AXL and MerTK, resulting in enhanced efferocytosis in TAMs. Proving the principle, we show that simultaneous blockade of CD276 and PD-1 restrain tumor growth better than any of the components as a single intervention. Taken together, our study supports a role for CD276 in efferocytosis by TAMs, which is potentially targetable for combination immune therapy.

Cancer immunotherapy embodies a strategy to combat malignancies by leveraging the cytotoxic potential of the host immune system, reflecting the importance of the interaction between the immune system and cancer. Immune checkpoint inhibitors, which can reinvigorate anti-tumor immune responses by disrupting co-inhibitory T-cell signaling, have revolutionized immunotherapy against various cancers over the last decade[1]. Since 2016, Food and Drug Administration (FDA) has approved several programmed cell death 1 (PD1)/PD1 ligand 1 (PD-L1) inhibitors for the clinical treatment of urothelial bladder cancer (BLCA), particularly for patients in advanced stages, who have a

[1]Department of Medical Oncology; Institute of Precision Medicine; Center for Translational Medicine, The First Affiliated Hospital of Sun Yat-sen University, Guangzhou 510080, China. [2]Department of Urology, Zhujiang Hospital, Southern Medical University, Guangzhou, China. [3]Institute for Advanced Study, Shenzhen University, Shenzhen 518057, China. [4]Hospital of Stomatology, Sun Yat-sen University, Guangzhou, China. [5]Senior Department of Oncology, the Fifth Medical Center of PLA General Hospital, NO.8 the east street, Fengtai District, Beijing 100071, China. [6]State Key Laboratory of Oral Diseases, National Clinical Research Center for Oral Diseases, Research Unit of Oral Carcinogenesis and Management, Chinese Academy of Medical Sciences, West China Hospital of Stomatology, Sichuan University, Chengdu 610041, China. [7]Department of Genetics, School of Life Sciences, Anhui Medical University, Hefei 230031, China. [8]These authors contributed equally: Maosheng Cheng, Shuang Chen, Kang Li, Ganping Wang, Gan Xiong. ✉e-mail: linshb6@mail.sysu.edu.cn; hao.xu@scu.edu.cn; chendm29@mail.sysu.edu.cn; liyang@ahmu.edu.cn; pengliang@301hospital.com.cn

median overall survival (OS) duration of 14.0 months and a 5-year survival of 13%[2]. Despite initial efficacy, durable responses are only achieved by a minority, approximately 15%–25%, with the majority experiencing relapse post anti-PD-1/PD-L1 therapy[3]. The mechanisms of immunotherapy resistance can be complicated and are incompletely understood. So far, several intrinsic and extrinsic mechanisms have been implicated to be in volved in mediating immunotherapy resistance, including genomic evolution of tumor cells, insufficient tumor antigenicity, alterations of immunosuppressive cells and cytokines, coinhibitory receptors and costimulatory receptors. These problems demand an intensive investigation into additional adaptive resistance mechanisms and rationale for targeting additional immune checkpoint axes in BLCA.

CD276 (B7-H3) is an immune checkpoint that belongs to the B7 immune co-stimulatory and co-inhibitory family. Protein level of CD276 is often elevated in tumor cells and stromal cells, while showing limited expression in normal tissues[4]. Moreover, overexpression of CD276 in tumor tissues frequently correlates with diminished tumor-infiltrating lymphocytes, hastened cancer progression, and adverse clinical outcome[5], which makes it an attractive and promising target for cancer immunotherapy. CD276-based immunotherapy strategies reported by our group and others, have demonstrated potent anti-tumor activity and feasibility for clinical application in solid tumors[6,7]. The antitumor effect of CD276 blockade is comprehensive, including the activation of T cell response, polarization of TAMs, induction of dendritic cells function, and inhibition of cancer-associated fibroblasts (CAF) activation. Furthermore, the administration of CD276 inhibitor suppresses tumor cell proliferation and invasion directly[5], indicating a potential of synergistic application with other immune checkpoint inhibitors. In BLCA, CD276 is also aberrantly expressed but the clinical significance is barely investigated[4,8]. However, little is known about the detailed role of CD276 in BLCA tumorigenesis and whether it could be an applicable intervention.

In this study, we demonstrate the critical roles of tumoral and TAM CD276 expression in driving BLCA development and progression. Our findings highlight the necessity of TAM CD276 expression for efficient efferocytosis and underscore its impact on TAM polarization towards a pro-tumoral phenotype. Depletion of CD276 leads to enhanced TAM MHCII expression, increased infiltration of cytotoxic CD8 + T cells, and reduced tumor growth in a mouse BLCA model, shedding light on the intricate mechanisms underlying tumor immune evasion. These results unveil the unanticipated role of CD276 in TAMs, presenting a compelling rationale for exploring CD276 blockade as a therapeutic strategy for bladder tumors.

## Results

### CD276 expression is a risk factor for prognosis in patients with BLCA

To investigate correlations between CD276 expression and BLCA patient survival, we first collected 93 tumor tissue specimens from BLCA patients who underwent surgical treatment at the Fourth Medical Center of PLA General Hospital and evaluated CD276 expression using immunohistochemistry (IHC) (Fig. 1a, Supplementary Data 1). Kaplan-Meier survival analysis and log rank tests showed that higher CD276 expression was associated with poorer overall survival (OS, $p = 0.0013$; Fig. 1b) and poorer progression-free survival (PFS, $p = 0.0004$; Fig. 1c). In addition, we examined the expression of CD276 in a commercial tissue microarray containing 63 BLCA and 16 para-tumor tissues via IHC staining (Supplementary Fig. 1a, Supplementary Data 2). Our results showed CD276 protein expression was upregulated in BLCA samples compared to para-tumor tissue samples (Supplementary Fig. 1b). Kaplan−Meier plot and log rank test showed that CD276 expression was positively associated with decreased OS ($p = 0.0022$; Supplementary Fig. 1c). To further validate our findings, The Cancer Genome Atlas (TCGA) BLCA cohort was used for analysis.

Similarly, mRNA expression level of CD276 was significantly higher in the BLCA tissues compared to the normal tissues (Supplementary Fig. 1d). In addition, patients with low CD276 mRNA expression level had better OS ($p = 0.0123$) as compared to patients with high CD276 mRNA expression level in the TCGA BLCA cohort (Supplementary Fig. 1e). Together, our data showed that CD276 is a marker for predicting prognosis in BLCA patients.

### CD276 global knockout leads to inhibition of murine BLCA development

To systematically test the function of CD276 in BLCA, we first generated CD276 whole body knockout (wKO) mice (Fig. 1d). Consistent with previous study[9], CD276 wKO mice were fertile and viable with normal life span. We then exposed paired cohorts of CD276 wKO and wildtype (WT) littermates to feeding water containing N-butyl-N-(4-hydroxybutyl) nitrosamine (BBN), a commonly used carcinogen for induction of rodent urinary bladder cancer that recapitulates human muscle-invasive bladder cancer[10](Fig. 1e). To examine the KO efficiency of CD276, we performed IHC staining of CD276. As illustrated in Fig. 1e, f, CD276 protein expression was virtually absent in CD276 wKO mice.

Notably, CD276 wKO led to significant improvement of survival for mouse bearing BLCA (Fig. 1g). In addition, the bladder indices, measured as ratios of bladder weight to body weight and BLCA tumor volume, were dramatically reduced after global KO of CD276 (Fig. 1h, i). Moreover, tumor burden of CD276 wKO mice was substantially lagged behind and reduced (Fig. 1j, k). We detected bladder cancer in the form of carcinoma in situ in WT mice as early as 16 weeks after BBN exposure. However, carcinoma in situ could only be detected in CD276 wKO mice after 24 weeks of carcinogen treatment (Fig. 1j, k). The incidence of cancer was also lower in CD276 wKO mice than in WT mice at 16, 20, 24 and 26 weeks (Fig. 1k). KI67 proliferation index was considerably lower in the CD276 wKO mice compared to the control group (Fig. 1l, m). In contrast, loss of CD276 resulted in the increased numbers of infiltrated CD8 T cells and levels of apoptotic cells in tumors (Fig. 1l, n, o).

### CD276 global knockout alters the cellular compositions and gene expression patterns of mouse BLCA

To comprehensively explore the underlying mechanism of CD276 in bladder carcinogenesis, we performed single-cell transcriptome profiling of the lesion foci from CD276 wKO ($n = 2$) and WT controls ($n = 2$). After rigorous quality control filtering, a total of 31,381 cells (10,550 and 20,831 cells for control and CD276 wKO tumors, respectively) were retained for subsequent analysis (Fig. 2a). The average number of unique molecular identifiers (UMI) per cell was about 3,530, and a median of approximately 1500 genes was detected per cell (Supplementary Fig. 2a, b). Based on the expressions of canonical markers, cells were divided into 7 major cell types, including epithelial cells (*Krt6a*, *Krt14*, *Krt17* and *Krtdap*), fibroblasts (*Col1a1*, *Col1a2* and *Col3a1*), endothelial cells (*Pecam1*, *Cdh5* and *Vwf*), T cells (*Cd3d*, *Cd3g* and *Trac*), dendritic cells (*H2-DMb2*, *H2-Oa* and *H2-Eb1*), neutrophils (*S100a8* and *S100a9*) and macrophages (*Apoe*, *Lyz2* and *C1qb*) by using uniform manifold approximation and projection (UMAP) (Fig. 2a, b). These cell types were consistently detected in all samples (Fig. 2c). In our datasets, epithelial cells were the most abundant cell type, contributing to almost 85% of total cells (Fig. 2d). We wondered whether mouse BCLA scRNAseq resembles those obtained from human BCLA, we then downloaded and analyzed an online human scRNAseq dataset, which includes 7 BLCA patient samples[11]. After the same quality control process, we obtained 34,022 cells, which were also divided into 7 major cell clusters, including epithelial cells, fibroblasts, endothelial cells, T cells, dendritic cells, neutrophils and macrophages (Supplementary Fig. 2c). Further analysis demonstrated that

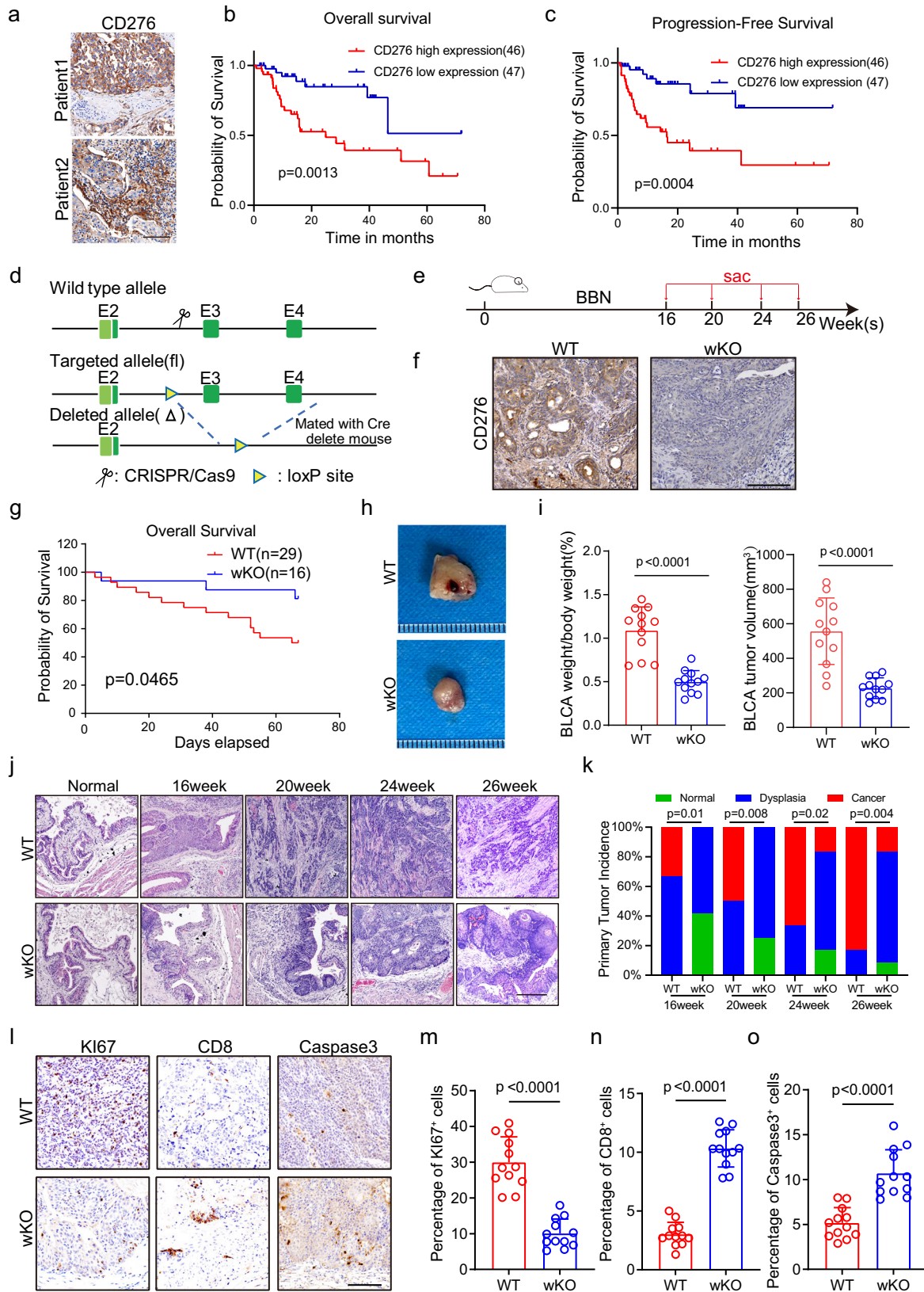

cell types and molecular signatures in mouse scRNA-seq datasets and online BLCA patient scRNA-seq dataset are highly consistent (Supplementary Fig. 2d, e).

To gain insights into how global CD276 depletion suppresses the progression of murine BLCA, we first analyzed the transcriptional and functional changes in epithelial cells. We subdivided epithelial cells

into six clusters (Supplementary Fig. 2f). Gene signature and pathway enrichment analysis showed that cluster 0 had higher expression of genes (e.g., *Jun, Cdkn1a and Hif1a*) in focal adhesion, glycolysis and Hif-1 signaling pathway; Cluster 1 upregulated genes (e.g., *Mki67, Top2a and Hmgb2*) involved in base excision repair, cell cycle and DNA replication; Cluster 2 had elevated expression of genes such as *Tjp1,*

**Fig. 1 | CD276 is highly expressed in BLCA and global knockout of CD276 can inhibit the progression of BLCA. a** Representative images of CD276 staining of human BLCA tissues from the Fourth Medical Center of PLA General Hospital cohort. Scale bar, 100 μm. **b, c** Kaplan-Meier survival curves were established by CD276 expression. Patients with BLCA were divided in high and low expression groups based on CD276 expression. Overall survival (OS, **b**) and progression-free survival (PFS, **c**) of PLAGH cohort are shown. *P* values were calculated by log-rank test. **d** Construction of Cd276 whole-body knockout (wKO) mice. **e** The experimental design of the bladder cancer tumorigenesis model and the schematic diagram of sample collection in batches. **f** Representative images of CD276 IHC staining in WT and wKO male mice. Scale bar, 100 μm. **g** The Kaplan-Meier overall survival curve of WT and wKO male mice. *P* value was calculated by log-rank test. **h** Representative image of BBN-induced bladder cancer. **i** Quantification of BLCA weight to body weight ratio (left) and BLCA tumor volume(right) in WT and wKO male mice. Data are presented as mean ± SD (*n* = 12). *P* value was calculated by two-tailed unpaired Student's *t* test. Representative images of H&E staining for BLCA at different time points (**j**) and quantification of primary tumor incidence (**k**) in WT and wKO male mice. Scale bar, 200 μm. *P* values were calculated by Pearson chi-square test. Representative IHC staining images (**l**) and percentages of KI67$^+$ (**m**), CD8$^+$ (**n**), and Caspase3$^+$ (Cleaved Caspase3, **o**) cells in WT and wKO male mice. Scale bar, 100 μm. Data were presented as mean ± SD (*n* = 12). *P* values were calculated by two-tailed unpaired Student's *t* test.

*G6pdx and Pgd*, which participate in adherent junction, glutathione metabolism and tight junction; Cluster 3 is characterized by expression of genes (e.g., *Ccnd1, Lamb3 and Lama3*) in p53 signaling pathway, focal adhesion and ECM-receptor interaction; Cluster 4 display high expression of *Gstm1, Idh1 and Mgst3* genes governing metabolic pathways such as fatty acid metabolism and glutathione metabolism; Cluster 5 showed increased expression of genes (e.g., *Atp6v1a, Ndufb7 and Ndufb10*) controlling oxidative phosphorylation and reactive oxygen species (Supplementary Fig. 2g, h). Besides, we also found Cluster 1 demonstrated highest cell cycle score while Cluster 3 are the major clusters display an EMT phenotype (Supplementary Fig. 2i). Remarkably, the proportions of both cell-cycling subclusters and EMT subclusters were decreased, while the proportion of Cluster 2 with high glutathione metabolism activity was increased in the CD276 wKO tumors as compared with control (Supplementary Fig. 2j). To identify genes regulated by CD276 in BLCA tumor cells, we performed differential expression analysis on epithelial pseudo-bulk transcriptomes from CD276 wKO and control samples. In total, we found 39 up-regulated genes and 99 down-regulated genes in CD276 wKO epithelial cells (Fig. 2e, Supplementary Data 3). Kyoto Encyclopedia of Genes and Genomes (KEGG) analysis showed that the downregulated categories related to focal adhesion, tight junction, PI3K-Akt signaling pathway and ECM-receptor interaction whereas upregulated ones were related to multiple metabolic pathways (Fig. 2f).

Broad expression of CD276 protein has been detected in human and mouse BLCA specimen (Fig. 1a, f). Besides, we found expression of CD276 in all cell types in mouse BLCA (Supplementary Fig. 2k). We wondered what is the functional role of stromal CD276 in BLCA. To test that, we first used CRISPR/Cas9 to delete CD276 in MB49 cells (CD276-SG), a mouse bladder carcinoma cell line, and validated CD276 KO by Western blotting (Supplementary Fig. 2l). We then subcutaneously injected parental or CD276-SG MB49 cells to WT or CD276 wKO mice. Our results showed depletion of CD276 in tumor microenvironment resulted in significant reduction of WT MB49 tumor growth and tumor size (Supplementary Fig. 2m–o), indicating that stromal CD276 is essential in BLCA development. Toward this end, we examined closely into changes in the tumor microenvironment (TME) components in our scRNAseq datasets. Noticeably, the composition of TAMs was drastically decreased in CD276 wKO mice as compared with that in WT mice while no notable change of the composition of other stromal cell types were found between WT and CD276 wKO mice (Fig. 2d). This inhibition of TAMs was validated by immunofluorescent (IF) staining of EMR1, a marker of TAMs, in BLCA samples (Fig. 2g). Given the CD276 was expressed in TAMs (Fig. 2g; Supplementary Fig. 2k), we wondered whether there is an autonomous effect of CD276 in regulation of TAMs in BLCA.

### Depletion of CD276 in TAMs suppresses the tumorigenesis of mouse BLCA

Hence, we generated transgenic mice carrying CD276 conditional knockout allele (*CD276$^{fl/fl}$*) (Fig. 3a). We crossed them with *LysM-Cre; Rosa-tdTomato* mice to obtain *LysM-Cre; Rosa-tdTomato; CD276$^{fl/fl}$* homozygous mutant (CD276 cKO) mice with myeloid-specific CD276

deletion. We first examined the effect of myeloid-derived CD276 on spleen in the healthy condition and found the ratio of spleen to body weight showed no significant difference between control and CD276 cKO mice (Supplementary Fig. 3a). Further analysis revealed no significant difference in percentage of macrophages, dendritic cells (DC) and myeloid-derived suppressor cells (MDSC) in spleen and bone marrow between control and CD276 cKO mice (Supplementary Fig. 3b, c).

The next step was to investigate the effect of myeloid-specific CD276 on growth of spontaneous BLCA. Control and CD276 cKO mice at six weeks of age were then exposed to BBN for 26 weeks (Fig. 3b). Initial analysis showed that CD276 cKO mice had lower bladder indices compared to the control mice (Fig. 3c). In addition, histological analysis revealed a lower incidence of invasive tumor formation in mutant compared to control mice (Fig. 3d). Results from IHC staining showed comparable levels of cell proliferation and an increase in apoptotic events in the invasive tumor cells of CD276 cKO compared to the control tumors (Fig. 3e, f). Furthermore, we detected more infiltrated CD8 T cells and less infiltrated TAMs in CD276 cKO mice compared to control mice (Fig. 3e, f).

To further characterize the cellular and molecular events in BLCA after CD276 cKO, we harvested lesion foci from CD276 cKO and control mice and performed scRNAseq. We were able to extract a total of 29,310 cells, with an average of 4123 UMI counts and 1783 detected genes per cell (Fig. 3g, i). Similar to the results stated above, we detected epithelial cells, fibroblasts, endothelial cells, T cells, dendritic cells, neutrophils and macrophages within these datasets (Fig. 3g, h).

Subsequently, we separated epithelial cells to examine the molecular events in tumor cells in CD276 cKO mice. After re-clustering epithelial cells, six major epithelial subpopulations were observed (Supplementary Fig. 3d). Each cluster was characterized by a specific gene signature, associated to distinct biological processes (Supplementary Fig. 3e, f). In particular, clusters 3 and 4 were remarkably decreased in CD276 cKO BLCA and displayed pronounced expression of EMT-related genes (Supplementary Fig. 3f, g). In agreement with our IHC results, we found that cluster 2, representing an active proliferating population, were comparable between CD276 cKO and control samples (Supplementary Fig. 3f, g).

### Expression of CD276 by TAMs promotes TAMs infiltration

To study the autonomous effects of CD276 on TAMs, we first looked into our scRNA-seq data. Our analysis revealed the transcriptional complexity of the TAMs population, which allowed us to identified three distinct TAMs clusters (Fig. 4a, b). KEGG analysis revealed diversity in biological pathways that were active within these clusters: a predominant cluster 0 characterized by activation of lysosome, phagosome, antigen processing and presentation; cluster 1 with increased activities of nucleotide metabolism phosphorylation and cell cycle and cluster 2 marked by enrichment of p53 signaling pathway, oxidative phosphorylation and glutathione metabolism (Fig. 4c). Although we found a markedly decrease in TAMs numbers in CD276 cKO samples compared to control, the ratios of each cluster remained comparable between CD276 cKO and control

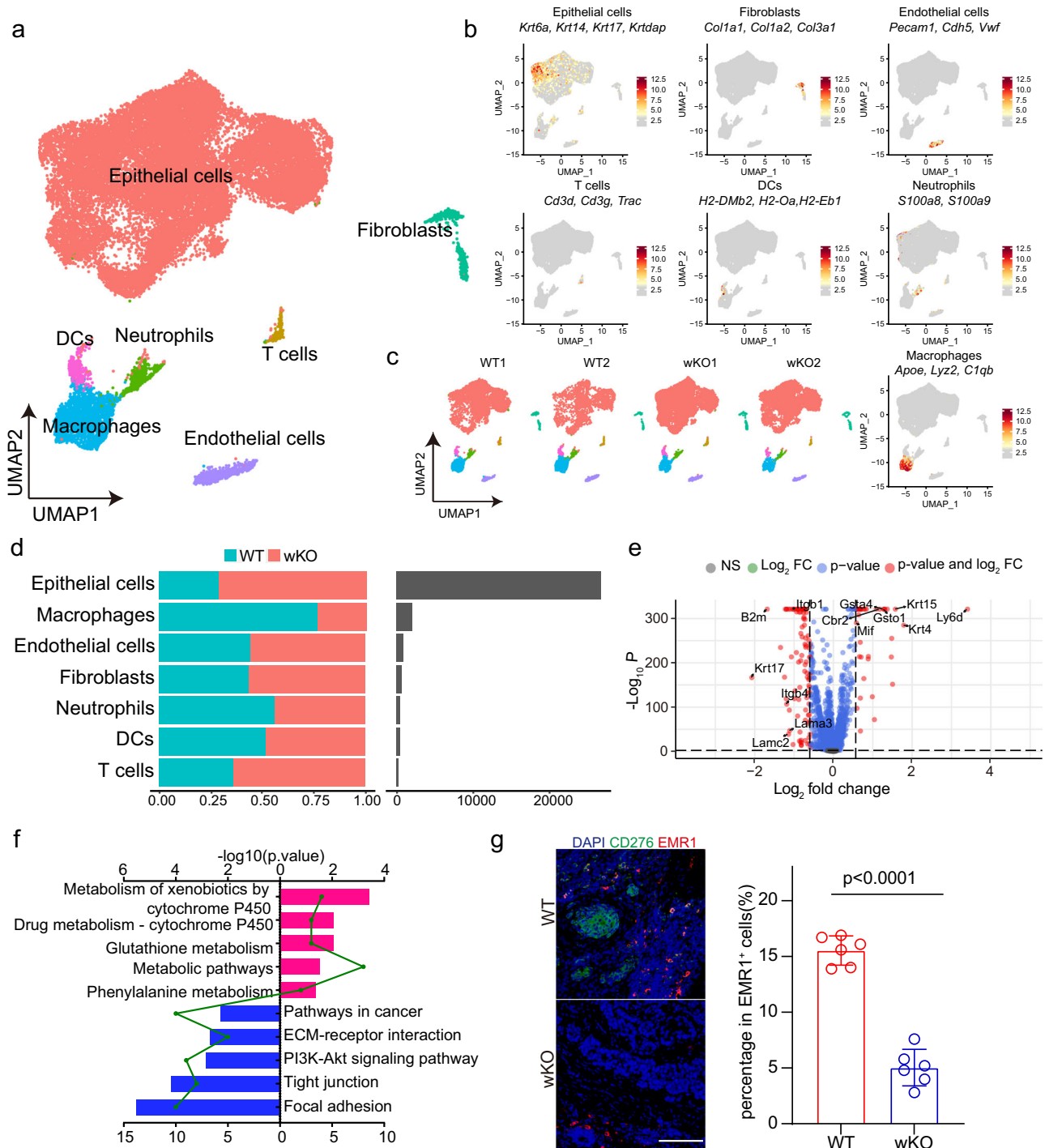

**Fig. 2 | Cd276 deletion leads to tumor landscape remodeling and inhibits TAMs infiltration in mouse bladder cancer. a** UMAP plot displaying the results after unbiased clustering. Subpopulations of epithelial cells, macrophages, endothelial cells, fibroblasts, neutrophils, DCs and T cells were identified, with each cell type colored. **b** UMAP plots showing the expression of feature gene sets in all cell type. **c** UMAP plots showing cell types in each bladder cancer sample. **d** Bar plots of proportion of cell type (left) and total cell number (right) in WT or wKO group.

**e** Volcano plot displaying the −Log10 P vs Log2 fold-change of genes differentially expressed between WT and wKO in epithelial cells. **f** KEGG pathway enrichment analysis using the intergroup differential gene of epithelial cells. **g** Opal/TSA multicolor IF staining with anti-CD276 and EMR1 antibodies (left). Nuclei are stained with DAPI (blue) and quantification of percentages of EMR1[+] cells (right) in WT and wKO male mice. Scale bar, 100 μm. Data was presented as mean ± SD ($n = 6$). $P$ value was calculated by two-tailed unpaired Student's $t$ test.

groups (Fig. 4d). Our flow cytometry analysis also revealed a significant reduction of macrophage infiltration in CD276-cKO mice compared to the control group. Notably, the infiltration ratio of MDSCs and DCs demonstrated no statistically significant alterations (Supplementary Fig. 4a–c). Interestingly, the proliferative and apoptotic ratios of TAMs were similar in CD276 cKO and control

groups (Supplementary Fig. 4d, e). Further analysis revealed that the ratio of M1 cells (*Nos2*, *Cd80* and *Cd86*)/M2 cells (*Mrc1*, *Arg1* and *Cd163*) was also not different between CD276 cKO and control samples (Supplementary Fig. 4f). Additionally, our immuno-fluorescence results showed no significant differences in macrophage M1 (INOS) or M2 (Cd163) phenotype activation within CD276

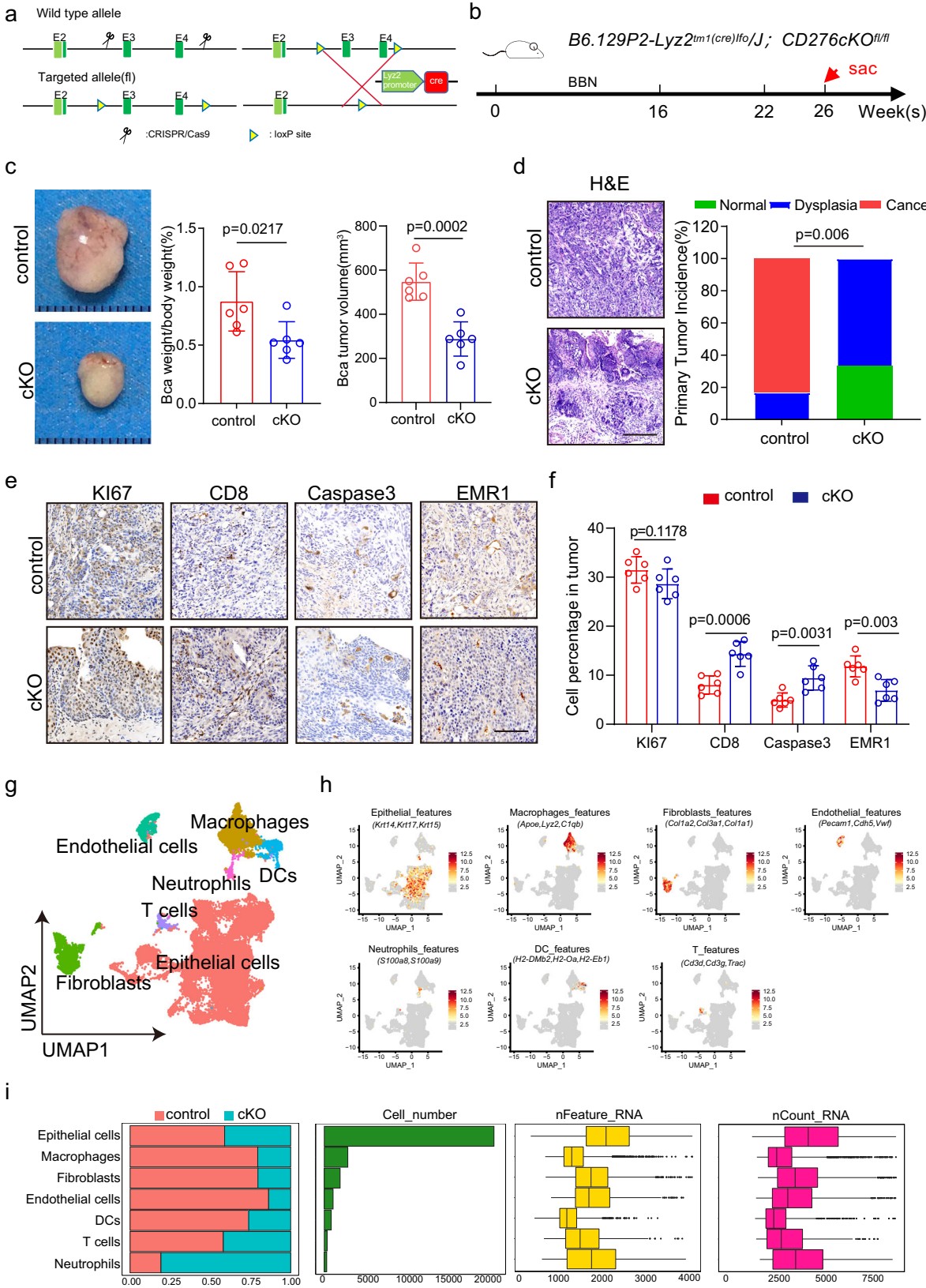

cKO tumors (Supplementary Fig. 4g)., suggesting that CD276 is dispensable for TAMs polarization in BLCA.

We then performed pseudobulk analysis to identify differentially expressed genes (DEGs) in TAMs after CD276 ablation and identified 331 upregulated genes and 492 downregulated genes in CD276 cKO TAMs compared to the control (Fig. 4e, Supplementary Data 4). In addition,

we tested for enrichment of genesets in KEGG databases. Our results showed strong downregulation of lysosome, endocytosis, phagosome and chemokine signaling pathways in CD276 cKO TAMs (Fig. 4f). Next, we determined the function of CD276 in regulating the expression of genes in the lysosome signaling pathway. Our results showed that CD276 deletion reduced the protein expression levels of CLTC (Clathrin

**Fig. 3 | Depletion of Cd276 in TAMs suppresses the tumorigenesis and remodels landscape of mouse BLCA. a** Construction of Cd276-conditional knockout (cKO) mice. **b** The experimental design of the bladder cancer tumorigenesis model. **c** Representative image of BBN-induced bladder cancer. Quantification of BLCA weight to body weight ratio and BLCA tumor volumes in control and cKO male mice. Data are presented as mean ± SD (n = 6). P value was calculated by two-tailed unpaired Student's t test. **d** Representative H&E staining of BLCA (left) and quantification of primary tumor incidence (right) in control and cKO male mice. Scale bar, 200 μm. P value was calculated by Pearson chi-square test. Representative images of KI67, CD8, Caspase3, EMR1 IHC staining (**e**) and percentages of KI67+,

CD8+, Caspase3+ and EMR1+ cells (**f**) in control and cKO male mice. Scale bar, 100 μm. Data are presented as mean ± SD (n = 6). P values were calculated by two-tailed unpaired Student's t test. **g** UMAP plot displaying the results after unbiased clustering. Subpopulations of epithelial cells, macrophages, endothelial cells, fibroblasts, neutrophils, DCs and T cells were identified, with each cell type colored. **h** UMAP plots showing the expression of feature gene sets in all cell type. **i** Data of the 7 subclusters of 29310 cells from 4 samples (from left to right): bar plots of proportion of cell type by control or cKO, and total cell number and box plots of the number of UMIs and genes.

Heavy Chain), LIPA (Lipase A, Lysosomal Acid Type), LAMTP5 (Lysosomal Protein Transmembrane 5) and LAMP2 (Lysosomal Associated Membrane Protein 2) in TAMs, which are key regulators of the lysosomal and phagocytic signaling pathways (Fig. 4g). Downregulation of chemokine signaling pathways might explain the reduced infiltration of TAMs in CD276 cKO mice (Fig. 4d). Our results further confirm that CD276 deletion reduces protein expression of chemokines including CCR1 (C-C Motif Chemokine Receptor 1), CCR5 (C-C Motif Chemokine Receptor 5), CX3CR1 (C-X3-C Motif Chemokine Receptor 1) and CXCR4 (C-X-C Motif Chemokine Receptor 4) in TAMs, which have been shown to promote macrophage infiltration[12–15](Fig. 4h). More importantly, recent studies have shown that genes involved in lysosome, endocytosis and phagosome pathway of TAMs led to clearance and degradation of apoptotic cells (efferocytosis) and subsequent inhibition of MHCII^high macrophages in cardiovascular system[16,17]. This prompted us to investigate whether ablation of CD276 in TAMs could induce MHCII expression. Indeed, our scRNAseq data showed MHCII feature genes (*H2-Ab1, H2-Eb1 and H2-Aa*) were significantly elevated in CD276 cKO TAMs (Supplementary Fig. 4h). Flow cytometry results further confirmed the percentage of TAMs with surface expression of MHCII protein was significantly elevated in CD276 cKO mice compared with controls, which was not observed in MDSCs and DCs (Fig. 4i, Supplementary Fig. 4i, j). Increase of MHCII expression often accompanies with recruitment of activated T lymphocytes. To prove this point, we collected MHCII+ TAMs and MHCII- TAMs using flow cytometry. We observed an increase in protein levels of chemokine CXCL9 (C-X-C Motif Chemokine Ligand 9) in MHCII+ TAMs, which has been confirmed to be related to T cell recruitment[18] (Fig. 4j). Our results showed a significant increase in the proportion of CD4+ T and CD8+ T cells in tumors after CD276 deletion (Fig. 4k). Detailed analysis of the T cells showed that Gzma+ CD8+ T cells were significantly increased in CD276 cKO tumors compared with control (Supplementary Fig. 4k, l). Moreover, IF staining of CD8 and GZMA and quantification results showed CD276 cKO tumors had a higher CD8+GZMA+ T cell proportion (54 ± 5.1%) compared to controls (20 ± 2.7%) (Fig. 4l). To examined whether activated T cells play a role in the anti-tumor effects of CD276 cKO, we block T lymphocytes following MB49 subcutaneous implantation by injection of anti-CD4 or anti-CD8 antibodies (Supplementary Fig. 4m). Profoundly, we found that depletion of either CD4+ or CD8+ T cells resulted in impaired of tumor suppressive effects of TAM CD276 cKO on BLCA (Supplementary Fig. 4m).

## CD276 mediates the expression of AXL/MerTK in TAMs via JUN

To comprehensively study the alterations of cell-cell communication after CD276 ablation in TAMs, we performed CellChat analysis[19] of our scRNAseq datasets to draw the full picture of intercellular signaling communications. Our results demonstrated complex cell-cell interaction networks between TAMs and other types of cells in both conditions (Fig. 5a, b). We found that the inferred number of interactions among BLCA is generally comparable between control and CD276 cKO samples. However, the interaction strength among cells was overall reduced in CD276 cKO tumors (Supplementary Fig. 5a). To be more specific and relevant, we focused on interactions between TAMs and other cells types. Interestingly, our comparison of the communication

patterns reflected a major alteration in TAMs-epithelial cells communication between the control and CD276 cKO sample (Fig. 5c, Supplementary Fig. 5b). We then predicted detailed receptor-ligand interactions between TAMs-epithelial cells using CellphoneDB[20]. Strikingly, we found GAS6-AXL, PROS1-AXL and GAS6-MerTK were most profoundly affected ligand-receptor paired interactions between TAMs-epithelial cells after depletion of CD276 in TAMs (Fig. 5d). The TAM receptors (AXL, MerTK and TYRO3) are tyrosine kinases which play critical roles in efferocytosis and establishing an immunosuppressive environment in tumor[21]. Both *Axl* and *Mertk* were significantly lower in TAMs after CD276 depletion based on our scRNAseq datasets, whereas *Tyro3* was not expressed in our data (Figs. 4e, 5d, Supplementary Fig. 5c). Furthermore, WB results showed AXL and MerTK protein expression was repressed after CD276 depletion in TAMs (Fig. 5g). Noticeably, *Axl* and *Mertk* also showed restricted expression in the TAMs in mouse BLCA (Supplementary Fig. 5d).

To study how CD276 can modulate the expression of AXL, we performed SCENIC analysis to determine the potential downstream master regulators of TAMs. In line with previous research[22], we found JUN was one of the most down-regulated master regulators in CD276 cKO TAMs compared to controls (Fig. 5e; Supplementary Fig. 5c, Supplementary Data 5), which was further validated by WB results (Fig. 5g). JUN has been known to be upstream of *Axl* and *Mertk* expression in cancer cells[23,24]. To investigate whether JUN can bind to the promoter of *Axl* and *Mertk* genes, we perform chromatin Immunoprecipitation quantitative real-time PCR (ChIP-qPCR) assay using antibodies against JUN or IgG (control). We observed enriched occupancy of JUN at the promoter regions of both *Axl* and *Mertk* genes compared to the IgG control, whereas JUN knockdown significantly reduced the binding of JUN to the promoter of *Axl* and *Mertk* in TAMs (Fig. 5f).

To examine whether CD276 is involved in efferocytosis, we co-cultured control or CD276-cKO TAMs, MDSCs and DCs with apoptotic MB49 that were labeled with the pH-sensitive dye pHrodo and measured efferocytosis. (Fig. 5h). Our results showed that in the CD276-cKO group, MDSCs and DCs had similar phagocytic levels compared to control, while the efferocytic of TAMs was significantly reduced (Fig. 5i–k). We wondered whether efferocytosis can affect the expression of MHCII in macrophages. We first examined the MHCII expression in bone marrow-derived macrophages isolated from control and CD276 cKO mice. Of note, CD276 cKO did not affect the expression of MHCII in bone marrow-derived macrophages (Supplementary Fig. 5e). However, we found similar inhibition of efferocytosis in bone marrow-derived macrophages after CD276 ablation when co-cultured with irradiated MB49 cells (Supplementary Fig. 5f). After incubation with MB49 cells, CD276 cKO macrophages showed significant up-regulation of MHCII expression (Supplementary Fig. 5g). To examine the role of JUN and AXL/MerTK in efferocytosis of TAMs, mice bearing BLCA were treated with either AP-1 inhibitor T-5224 or AXL/MerTK inhibitor R428. Despite the infiltration of TAMs was unaltered in tumors treated with either T-5224 or R428 (Supplementary Fig. 5h, j), the efferocytosis of TAMs were significantly inhibited by T-5224 or R428 treatment in vitro (Fig. 5l). In addition, MHCII expression in TAMs was dramatically elevated in TAMs treated with T-5224 or R428 (Supplementary Fig. 5i, k).

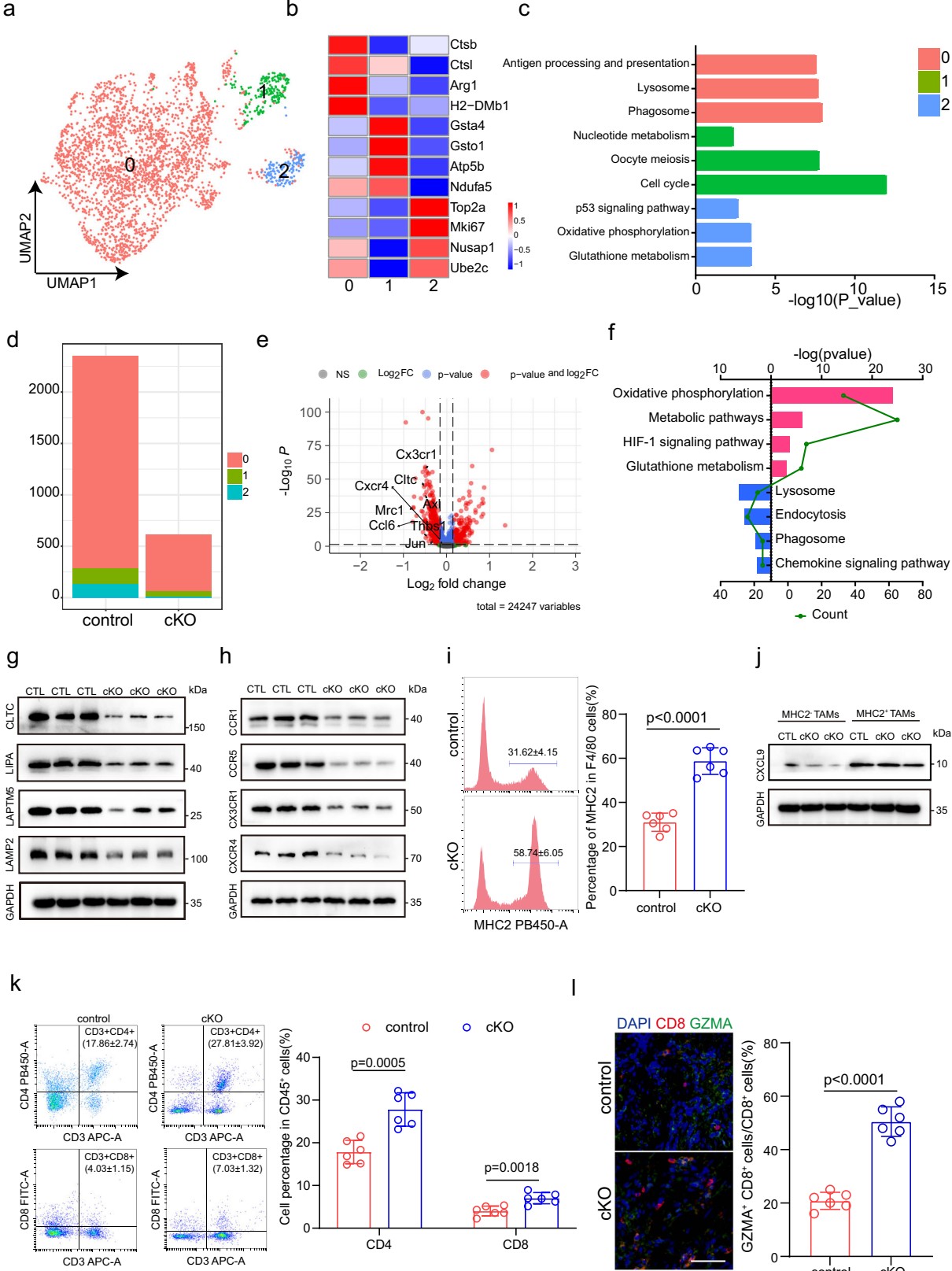

## Overexpression of CD276 in TAMs promotes efferocytosis and BLCA progression

To further validate our findings, we generated a CD276 conditional knock-in transgenic mice (*ROSA-CD276*), in which the murine CD276 cDNA sequence was inserted into the Rosa26 locus downstream of a floxed transcriptional stop cassette, thereby allowing for Cre-dependent overexpression of full-length CD276 (Fig. 6a). We then crossed *ROSA-CD276* mice with *LysM-Cre* mice to obtain *LysM-Cre; ROSA-CD276* (CD276 cKI) mice with myeloid-restricted CD276 overexpression. Western blot analysis confirmed overexpression of CD276 in the macrophages of CD276 cKI mice (Supplementary Fig. 6a). Under physiological conditions, neither CD276 overexpression nor treatment

**Fig. 4 | Depletion of Cd276 in TAMs alters the cellular compositions and gene expression patterns. a** UMAP plot showing 3 clusters of 2967 macrophages (indicated by colors). **b** Heatmap of signature genes for macrophages clusters. Each cell cluster is represented by four specifically expressed genes. **c** KEGG pathway enrichment analysis using the characteristic genes of macrophage subpopulation. **d** Bar plots of number of the subclusters of macrophages in control and cKO male mice. **e** Volcano plot displaying the −Log10 P vs Log2 fold-change of genes differentially expressed between control and cKO in macrophages. **f** KEGG pathway enrichment analysis using the intergroup differential gene of macrophages. **g** Western blot of CLTC, LIPA, LAPTM5 and LAMP2 in macrophages from control and cKO groups using GAPDH as loading control. **h** Western blot of CCR1, CCR5, CX3CR1 and CXCR4 in macrophages from control and cKO groups using GAPDH as loading control. **i** Representative flow cytometry plots (left) and statistical analysis of MHCII⁺ macrophages (right) in control and cKO groups. Data are presented as mean ± SD ($n = 6$). $P$ value was calculated by two-tailed unpaired Student's $t$ test. **j** Western blot of CXCL9 in MHCII⁺ TAMs and MHCII⁻ TAMs using GAPDH as loading control. **k** Representative flow cytometry plots (left) and statistical analysis of CD3⁺CD4⁺ T cells and CD3⁺CD8⁺ cells (right) in control and cKO groups. Data are presented as mean ± SD ($n = 6$). $P$ value was calculated by two-tailed unpaired Student's $t$ test. **l** Representative immunofluorescence (IF) staining images of CD8 (red) and GZMA (green, right). Statistical analysis of the ratio of CD8⁺ GZMA⁺ cells to CD8⁺ cells (CD8T killing capacity) in control and cKO male mice (left). Scale bar, 50 μm. Data are expressed as mean ± SD ($n = 6$). $P$ values were calculated by two-tailed unpaired Student's $t$ test.

with R428 elicited alterations in the ratio of spleen to body weight or the proportion of EMR1⁺ macrophages in the spleen or bone marrow. (Supplementary Fig. 6b–e).

Next, we examined whether CD276 overexpression could promote BLCA development in mouse and whether treatment of R428 could lead to reversion of the phenotype (Fig. 6b). We found that R428 improved the survival of control and CD276 cKI mice at 26 weeks after initial BBN treatment (Fig. 6c). CD276 cKI mice developed invasive carcinoma, whereas no invasive carcinoma was observed in control or CD276 cKI mice treated with R428 (Fig. 6d–h). In addition, CD276 cKI mice displayed higher bladder index compared to control, whereby increase bladder index was abrogated by treatment of R428 (Fig. 6e, f). We detected lower apoptotic rates in the invasive tumor cells of CD276 cKI tumors compared to the control tumors (Fig. 6i, j). On the contrary, R428 led to increased level of cell apoptosis in both control or CD276 cKI tumors (Fig. 6i, j). Of note, despite CD276 overexpression in TAMs resulted in increase of EMR1⁺ macrophages in BLCA tumors, no overt difference was observed in EMR1⁺ macrophages recruitment treated with R428 in control or CD276 cKI mice (Fig. 6k, l). Moreover, TAM CD276 overexpression-mediated efferocytosis was abrogated by R428 treatment (Fig. 6m, n), while TAM CD276 overexpression-mediated inhibition of MHCII was rescued by R428 treatment (Fig.6o, p). Concurrently, the inhibition of CD4⁺ T, CD8⁺ T and GZMA⁺CD8⁺ T cells by TAM CD276 overexpression were reversed by R428 treatment (Fig. 6q, r, Supplementary Fig. 6f, g). Similarly, we were able to observe enhanced BLCA growth in CD276 cKI mice using a MB49 transplantation BLCA model (Supplementary Fig. 6h–j). At the same time, we observed less CD8⁺ and GZMA⁺CD8⁺ T cells in MB49 tumors from CD276 cKI mice compared to control sample (Supplementary Fig. 6k–m). Consistently, treatment of R428 was able to increase CD8⁺ and GZMA⁺CD8⁺ T cells in MB49 tumors (Supplementary Fig. 6k–m).

### CD276 enhanced activity in combination with PD-1 blockade in murine BLCA

Given ablation of CD276 in TAMs promotes the expression of MHCII and recruitment of cytotoxic T cells. We hypothesized that combination treatment with anti-PD-1 antibody would thereby increase the capability of anti-CD276 antibody to elicit anti-tumor activity. To test that, we treated mice bearing BLCA tumors with IgG, anti-CD276 antibody, anti-PD-1 antibody or combination of anti-CD276 and anti-PD-1 antibodies (Fig. 7a). We discovered a noteworthy enhancement in the survival duration of BLCA mice through the implementation of single and combined therapeutic approaches (Supplementary Fig. 7b). Our findings demonstrated that both individual treatment agents and the combined therapy contributed significantly to the reduction of tumor growth and bladder index. Particularly, the combination therapy outperformed the effects of the individual treatment agents, showcasing its superior efficacy. (Fig. 7b–d). Histological examination revealed that the tumor incidence was lowest in combined therapy group than other groups (Fig. 7e. f). Moreover, compared with other treatment groups, the combined treatment group led to an increase in the level of apoptosis (Fig. 7g, h). IF staining revealed that anti-CD276

treatment or combined therapy significantly decreased the frequencies of EMR1⁺ macrophages (Fig. 7i, j). In addition, we found treatment of anti-PD-1 antibody did not dramatically alter the MHCII expression and efferocytosis in TAMs (Fig. 7k–n). Intriguingly, we observed that anti-CD276 and anti-PD-1 synergistically increased the proportion of infiltrated CD4⁺ T, CD8⁺ T and CD8⁺GZMA⁺ T cells (Fig. 7o, p, Supplementary Fig. 7d–g). In order to further demonstrate the efficacy of the combination therapy, we performed two additional tumor transplantation experiments in mice in which MB49 was injected into C57 mice and MBT2 was injected into C3H mice (Supplementary Fig.7a). The critical value of tumor size was used as the sampling endpoint until day 60. Our results show that combined therapy could improve survival in tumor-bearing mice compared to IgG or single-dose therapy (Supplementary Fig.7c). These data indicate a combination of anti-PD-1 and anti-CD276 therapy could optimally induce effective anti-tumor immunity in mouse BLCA.

## Discussion

Application of immune checkpoint inhibitors has revolutionized the treatment of various solid malignancies. However, the clinical efficacy of immune checkpoint therapy is limited to a subset of patients with specific tumor types[25]. Due to the tumor heterogeneity and the complexity of tumor microenvironment, combinatorial immune checkpoint blockade strategies are considered promising to overcome this problem[26]. However, the mechanistic rationale for the combination of immune checkpoints is elusive. In addition to the well-studied PD-1/PD-L1 and CTLA4, CD276 is also a member of immune checkpoints and the blockade of which are proved effective in the treatment of many solid tumors. CD276 protein is broadly expressed in tumor lesion, including tumor cells, dendritic cells and macrophages, which makes CD276 blockade will lead to comprehensive consequences. In the current study, we decipher the function of CD276 by genetic ablation and following single cell transcriptome analysis in murine BLCA model, which help to elucidate role of CD276 in cancer immune responses and provide evidence of synergistic anti-tumor effect of CD276 and PD-L1 blockade in cancer treatment.

Despite there are still challenges regarding identify the receptor(s) of CD276 and its specific roles in different cells and complicated immune responses[6], multiple approaches targeting CD276, including blocking mAbs[27], bispecific mAbs[28–31], chimeric antigen receptor T cells[32–34], and combination therapies[9,35], have been tested and proved to be promising in the treatment of various solid tumors. A better understanding of CD276 functions in the context of tumor microenvironment will undoubtedly define the applications of CD276 blocked in a more precision way. Previous investigations of CD276 mainly focused on its role in tumor cells. As a member of B7 family which consists of immune regulatory ligands, CD276 could modulate T lymphocyte activation and differentiation, recent studies have revealed that CD276 induces a robust immune evasive effect when deregulated in cancers. In addition, CD276 is thought to have non-immunological roles in cancer progression by activation of proliferation, invasion, metastasis, and resistance to chemotherapy[4].

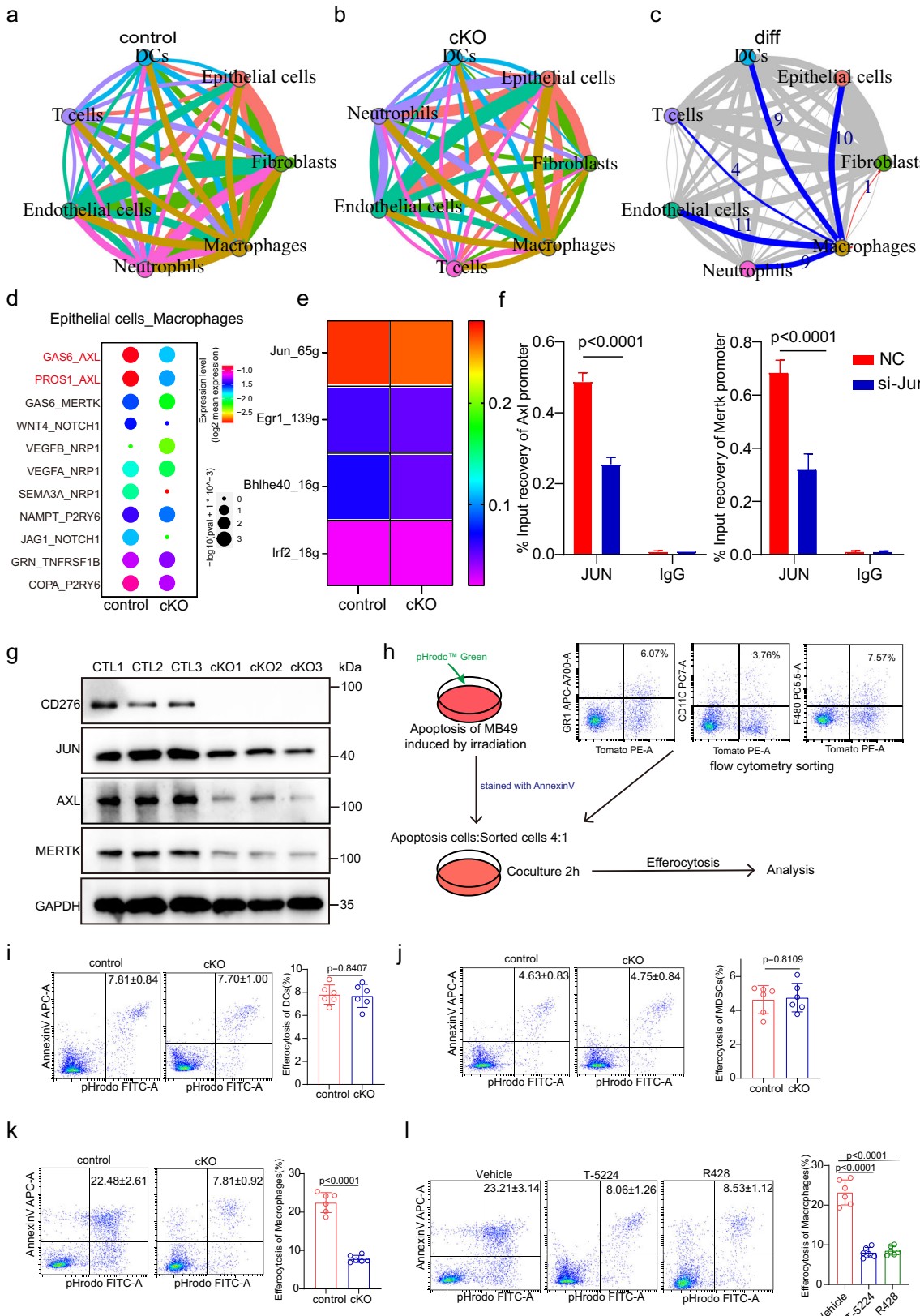

Augmented expression of CD276 is also observed in tumor blood vessels[27], indicating its angiogenesis function as treatment target as well. In the current study, our single cell sequencing data revealed that CD276 mainly expressed in tumor cells, endothelial cells and TAMs. Whole body KO of CD276 lead to a significant reduction of macrophage cells in tumor tissue rather than in spleen or bone marrow, this

might because normal macrophages were only induced to express CD276 by tumor cells[36], and our transcriptome analysis indicates that chemotaxis/migration signaling were also downregulated upon CD276 knockout in macrophages. More importantly, ablation of CD276 in TAMs is sufficient to inhibit the BLCA progression through mediating efferocytosis. Considering the efferocytic clearance of apoptotic cells

**Fig. 5 | CD276 promotes efferocytosis by regulating expression of AXL/MerTK.**
CellphoneDB results showing the capacity of intercellular communication in control (**a**) and cKO (**b**) BLCA tissues (the width of the line is determined by the number of interactions between cell types and the color of the line is determined by the transmitter). **c** Network plot showing the difference in the capacity of intercellular communication between control and cKO BLCA tissues (blue line represents downregulation of intercellular communication, red represents upregulation, and the width of the line represents the number of up- or down-regulations). **d** Dot plot showing the expression of receptor-ligand pairs between macrophages and epithelial cells in control and cKO BLCA tissues (colors represent mean expression of receptors and ligands). **e** Heatmap showing the activity of TFs in macrophages from control and cKO male mice. **f** ChIP-qPCR analysis of JUN binding to Axl and Mertk genes. Data are presented as mean ± SD (*n* = 3). *P* values were calculated by two-tailed unpaired Student's *t* test. **g** Western blotting of CD276, JUN, AXL, MERTK and GAPDH in control and cKO BLCA macrophages. **h** Flow chart of efferocytosis. The apoptotic cells induced by irradiation were co cultured with the sorted cells for 2 h and then analyzed by flow cytometry. Representative flow cytometry plots and quantification of percentage efferocytosis in DCs (**i**), MDSCs (**j**) and macrophages (**k**) from control and cKO male mice. Data presented as mean ± SD (*n* = 6). *P* values were calculated by two-tailed unpaired Student's *t* test. **l** Representative flow cytometry plots of tumor (left) and quantification of percentages of efferocytosis in macrophages of different treatment groups (right). Data was shown as mean ± SD (*n* = 6). *P* value presented by one-way ANOVA with Tukey's multiple comparison test.

by TAMs promotes the resolution of inflammation by both avoiding secondary inflammatory cell death and altering phagocyte priming[37], our results indicate the essential oncogenic roles of CD276 in TAMs.

The molecular signaling related to CD276 in cancer immune response is not fully understood, especially the downstream functional alteration and associated pathways upon CD276 inhibition. Our results suggest that CD276 activates JUN to promote the expression of AXL/MerTK in TAMs, which in turn induces the efferocytosis of TAMs and inhibits the MHCII mediated antigen presenting of TAMs. It has been reported that blocking apoptotic cell clearance by targeting MerTK of TAMs will stimulate T cell activation and synergize with anti-PD-1 or anti-PD-L1 therapy[38]. In addition, inhibition of AXL in TAMs could also stimulate the immune response in acute leukemias[39], which suggested AXL is also a potential target that plays vital roles in myeloid-centered immunotherapy. Our results showed clearly that knock out CD276 lead to a broad alteration of associated signaling and tumor promoting roles in TAMs, including downregulations of MerTK/AXL and efferocytosis, indicating a broader application scenario in cancer immunotherapy and lower possibilities of drug resistance.

In sum, we report that both mouse and human TAMs in BLCA express high levels of CD276. CD276 expression on TAMs is necessary and sufficient to drive the progression of BLCA. Our study expands the role of CD276 in myeloid subsets of TME, suggesting that CD276 exerts its immunosuppressive function across the innate and adaptive immune system. In addition, upregulated expression of CD276 in TAMs could be assessed in future investigation as a potential contributor to the treatment efficacy of anti-CD276 blockade therapy.

## Methods
### Ethics statement
All animal experiments described in this study were reviewed and approved by the Animal Care and Use Committee of Sun Yat-sen University (SYSU-IACUC-2021-000138). The Committee limits tumor growth to no more than 10% of the animal's original body weight and the average tumor diameter to no more than 20 mm. Tumor samples were collected with the patients' written informed consent and approved by the Human Research Ethics Committee of the Fourth Medical Center of PLA General Hospital. Human bladder cancer tissues arrays are purchased from Shanghai OUTDO Biotech Co., Ltd.

### Mice, bladder cancer models and drug treament
WT C57BL/6(BCG0206), *Cd276*[-/-](GJ-077-1), *Cd276*[fl/fl] (GJ-077-2) and *Rosa-Cd276* (CYX-077) mice were purchased from The Biocytogen (Nantong, China). *Rosa-tdTomato* (007914) mice were purchased from The Jackson Laboratory (Bar Harbor, ME, USA). C3H/He (BCG0205) mice were purchased from The Biocytogen (Nantong, China). *Axl*[-/-] mouse strain was a gift from Dr. Guoqiang Yuan (The Second Hospital of Lanzhou University). Lyz2-Cre (*B6.129P2-Lyz2*[tml (cre) lfo/J]) mouse were kindly given by Dr. Xiaojun Xia (Sun Yat-sen university Cancer Center). All animals in this study were maintained under specific pathogen free conditions and housed under a 12-h light/dark cycle and given ad libitum access to food and water. For bladder cancer induction, male mice at least 6 weeks old received BBN at a dose of 0.05% in drinking water as previously describe[10]. The mice used in the study were all male.

T-5224 (HY-12270, MCE, Shanghai, China) was administered orally to male mice at a dose of 600 μg per 20 g body weight three times a week for 2 weeks. R428 (S2841, Selleck, Shanghai, China) was administered orally to male mice at a dose of 100 μg per 20 g body weight twice a day. For CD276 and PD1 blockade treatment, anti-CD276 antibody (InVivo-MAb anti-mouse CD276, BE0124, BioXcell 10 mg/kg body weight) was administrated three times a week for 4 weeks and anti-PD1 (BE0146, BioXcell, Shanghai, China) was administrated i.p. at a dose of 200 mg/mouse at day 1, 3, 5, 7 and 14. Anti-CD8α (InVivoPlus anti-mouse CD8α, BP0061 BioXcell, 100 μg/mice) and anti-CD4 (InVivo-Plus anti-mouse CD4, BP0003-3 BioXcell, 100 μg/mice) were administrated twice a week for 2 weeks. Mice injected with vehicle or IgG isotype antibody were used as control.

### Tissue preparation, cell isolation and scRNA-seq
After 26 weeks of exposure to BBN, control, *Cd276*[-/-] and *Lyz2-Cre; Cd276cKO*[fl/fl] male mice were euthanized by CO$_2$ with secondary cervical dislocation. The bladder was dissected and the tumor collected. The tumor samples were gently minced into 1 mm$^3$ pieces and digested with mouse Tumor Dissociation Kit (130-096-730, Macs Miltenyi Biotec, Guangzhou, China) at 37 °C for 45 min using gentleMACS Dissociator (130-093-235, Macs Miltenyi Biotec, Guangzhou, China). Reaction was deactivated by adding FBS to 10%, then solution was passed through a 40 μm Falcon cell strainer (352340, Corning, Guangzhou, China). After centrifugation at 500 × g for 5 min, cell pellet was incubated with 1 mL of ACK Lysing Buffer (A1049201, Thermo Fisher Scientific, Shanghai, China) on ice for 5 min to get rid of red blood cells. Dead cells were removed using the MS columns of the Dead Cell Removal Kit (130-090-101, Macs Miltenyi Biotec, Guangzhou, China) following the manufacturer's protocol. Live cells were resuspended in PBS with 0.04% BSA and counted using a Countess 3 Automated Cell Counter (Thermo Fisher Scientific, Shanghai, China). The GEXSCOPER microfluidic chip is used to capture single cell suspensions at concentrations of 1 to 3 × 10$^5$ cells/mL according to the Singleron GEXSCOPER procedure. Each microwell contains a magnetic bead paired with a cell, each with a unique cell label and multiple molecular labels [Unique Molecular Identifiers (UMIs)] used to capture RNA so that it can hybridize with the beads when RNA is released from the captured cells. The magnetic beads are then collected into 1.5 ml Eppendorf tubes for reverse transcription and cDNA synthesis is performed on single cells using the GEXSCOPER Single Cell RNA Library Kit[40]. Following cDNA synthesis and PCR amplification, cDNA fragment size was determined using Qubit (Thermo, Waltham, MA, USA) for cDNA quantification and a fragmentation analyser. Libraries were constructed after a series of steps including fragmentation, adapter ligation, purification, PCR amplification, size selection and quality control. The libraries were finally sequenced on the Illumina HiSeq X platform using 150 bp paired-end reads.

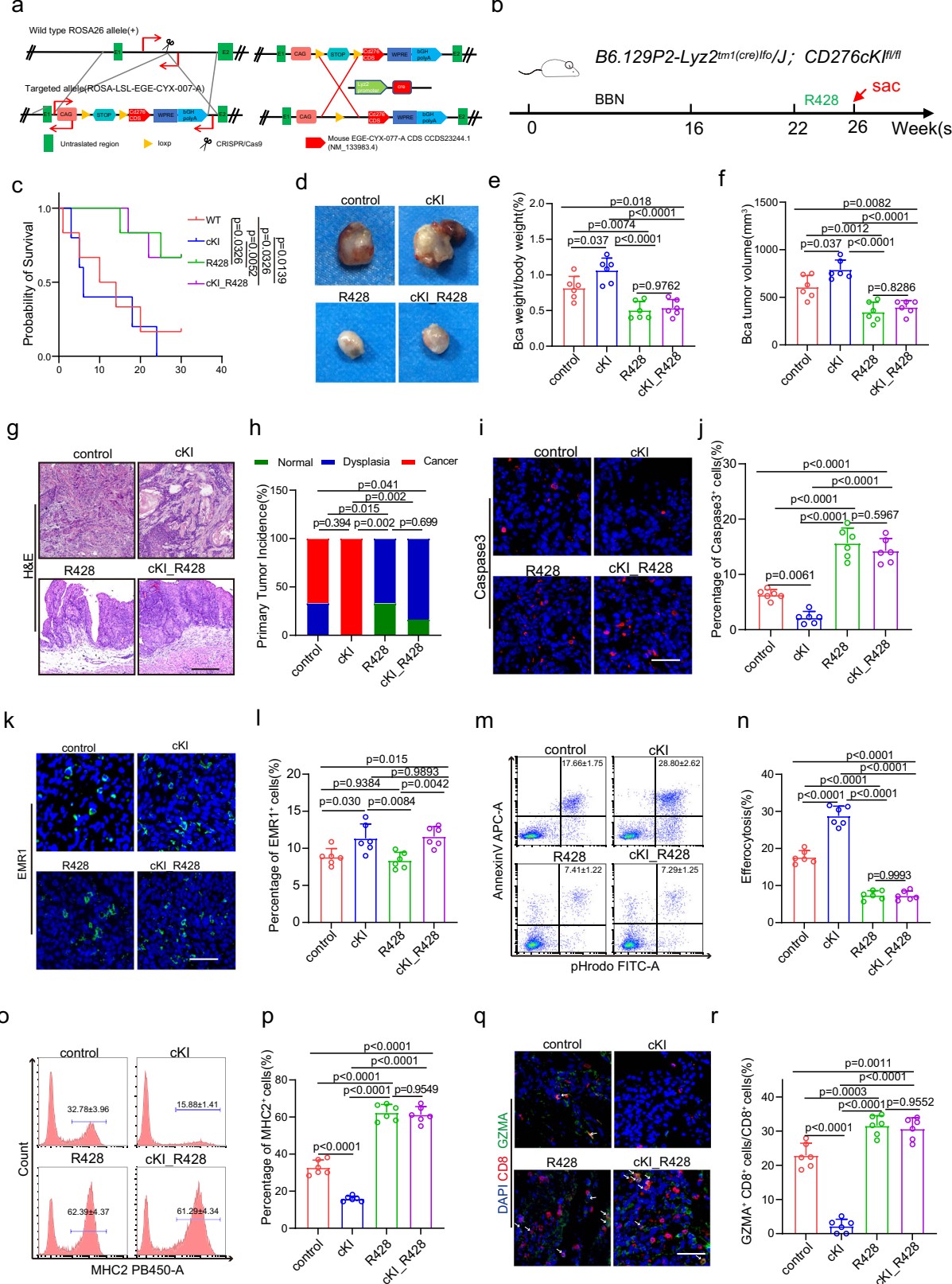

## Sequencing data processing

We used CeleScope (https://github.com/singleron-RD/CeleScope) software created by Singleron for generation of gene expression matrices from the raw data. Briefly, reads were compared to the reference genome GRCh38 with the ensemble version 93 gene annotation using STAR following quality control and filtering of the data using fastqc (version 0.11.7) and cutadapt (version 1.17, version 2.6.1b). Then, the gene count matrix was produced using feature-Counts (version 1.6.2). Finally, the gene expression matrix was imported into the R (version 4.0.0) software and the Seurat (version 4.0.0) package was used for the downstream analysis. Genes found in less than three cells in each cell were filtered. Additionally, cells

**Fig. 6 | Treatment of R428 reversed Cd276 overexpression-mediated phenotype. a** Construction of CD276-conditional knock-in (cKI) mice. **b** The experimental design of the bladder cancer tumorigenesis model and strategies of treatment. **c** The Kaplan-Meier overall survival curve in different treatment groups. *P* values were calculated by log-rank test. **d–f** Representative image of BBN-induced bladder carcinoma (**d**). Quantification of BLCA weight/body weight ratio (**e**) and BLCA tumor volume (**f**) in different treatment groups. Data are presented as mean ± SD (*n* = 6). *P* values were presented by one-way ANOVA with Tukey's multiple comparison test. Representative H&E staining of BLCA (**g**) and quantification of primary tumor incidence (**h**) in different treatment groups. Scale bar, 200 μm. *P* values were calculated by Pearson chi-square test. **i–l** Representative immunofluorescence (IF) staining images of Caspase3 (**i**) and EMR1 (**k**). Quantification of the percentage of Caspase3⁺ (**j**) and EMR1⁺ (**l**) cells in different treatment groups. Scale bar, 50 μm. Data are presented as mean ± SD (*n* = 6). *P* values were presented by one-way ANOVA with Tukey's multiple comparison test. Representative flow cytometry plots (**m**, **o**) and statistical analysis of percentages of efferocytosis (**n**) and MHCII⁺ cells (macrophage antigen presenting capacity, **p**) in macrophages from different treatment groups. Data are presented as mean ± SD, *n* = 6. *P* values were presented by one-way ANOVA with Tukey's multiple comparison test. Representative immunofluorescence (IF) staining images of CD8 and GZMA (**q**) and statistical analysis of the ratio of CD8⁺ GZMA⁺ cells to CD8⁺ cells in different treatment groups (**r**). Scale bar, 100 μm. Data was shown as mean ± SD, *n* = 6. *P* values were presented by one-way ANOVA with Tukey's multiple comparison test.

with less than 200 and more than 5000 genes as well as cells with higher than 10% expressions of the mitochondrial or hemoglobin genes were eliminated. Using the default parameters of the DoubletFinder R package (version 2.0.3), doublets were excluded. After quality control process, the count data were first normalized using the NormalizeData function. The expression data of all genes were then normalized using the ScaleData function, 2000 intercellular variant genes were chosen using the FindVariableFeatures function, and these variant genes were then mapped to the low-dimensional subspace using the RunPCA principal component analysis tool. In the 10 principal component low-dimensional subspaces, common nearest-neighbor graphs were built using the FindNeighbors function and the Euclidean distance metric. The RunUMAP function was used to visualize the clustered cells after the cells had been clustered using the FindClusters function. At the same time, we downloaded online available scRNAseq datasets from Gene Expression Omnibus (GSE135337), including from 7 human bladder cancers. All quality control, normalization, and downstream analysis was performed using the Seurat unless otherwise noted.

## Cell type annotation and Identifying maker genes

Integrated data was then subjected to nonlinear dimensionality reduction, clustering, and visualization[41,42]. To identify marker genes, the FindAllMarkers function was used with likelihood-ratio test for single cell gene expression. The average expression of the markers inside each cluster was utilized to represent the data on a heatmap. The FindIntegrationAnchors and IntegrateData functions were used to find anchors and compare the cell types of distinct samples. The FindMarkers function was used to identify the genes that were differentially expressed for various groups.

## SCENIC analysis

The gene regulatory networks in the scRNA-seq data were investigated using the R package SCENIC (version 1.2.1)[43]. The raw count matrix of the combined data set, which contained both control and cKO, was used as input after low-quality cells were removed. The default settings were left in place for analysis. RcisTarget found possible regulons by searching the SCENIC-provided mm10 motif ranking database for the mouse species. Transcription factor modules that had more than 10 genes mapped to them were kept for additional study (annotations with the suffix "_extended" indicated "low confidence").

## Comparison of human and mouse bladder cancer cell types Heatmaps and the Sankey diagram

We used GSE135337 dataset to see if the cell types that identified from mouse bladder samples are comparable to those from human samples. First, we discovered genes that are orthogonal between mice and humans. Second, we combined cells from two datasets into a single space using the mutual closest neighbors approach. Using Seurat's sctransform function, we identified the most variable genes in each dataset. Next, we used the RunFastMNN function (SeuratWrappers, version 0.1.0) to generate an integrated cell matrix. Third, we used the

Seurat package to perform clustering and cell-type identification. The Sankey diagram was then created after we had examined the origin of each cell type[44]. We compared mouse and human bladder cancer cell types systematically at the single cell and whole transcriptome levels in order to define how bladder cancer cell types relate to one another across species. We then created the mouse-human bladder cancer cell types comparison dendrograms and heatmaps as previously described[45].

## CellChat and CellPhoneDB analysis

CellChat v1.1.0 (github.com/sqjin/CellChat) was used for cell-cell interaction analysis and visualization. The cell type labels utilized were determined from the Harmony integration findings utilizing all single-cell data sources. For each step's parameterization, default values were used. We used CellPhoneDB (version 2.1.7) to analyze our data and deduce the presence of ligand-receptor interactions in WT and cKO cells utilizing the package's default pipeline implementation. Using Biomart, the genes from mice were converted into those from humans. Non-log-transformed UMI counts were used to calculate the expression levels of receptors and ligands[20].

## Histology and immunohistochemistry

The mouse bladder cancer tissues were fixed in neutral formalin for 48 h, embedded in paraffin, and sectioned into 5 μm pieces. Dewaxing hydration was used to wash away the paraffin from the slices. Utilizing Sodium Citrate Antigen Retrieval Solution, antigen retrieval was carried out (C02-02002, Bioss Antibodies, Beijing, China). By soaking the slides in 3% hydrogen peroxide (CS7730, G-clone, Beijing, China) for 10 m, the endogenous peroxidase activity was inhibited. The sections were then blocked for 30 m at 37 °C in blocking solution containing 5% BSA (SW3015, Solarbio, Beijing, China) and 0.1% Triton X-100 (Boster Biological Technology Co., Ltd., Wuhan, China). CD276 (14058, CST, Guangzhou, China, 1:200), MKi67 (NB500-170, Novus, Guangzhou, China, 1:300), CD8 (ab52625, Abcam, Guangzhou, China, 1:200), EMR1 (27044-1-AP, Proteintech, Wuhan,China, 1:200), GZMA (A6231, abclonal, Guangzhou, China, 1:200) and Cleaved Caspase-3 (9661 S, CST, Guangzhou, China, 1:200) primary antibodies were incubated on section for an overnight period at 4 °C. Following that, each segment was incubated with a suitable secondary biotinylated antibody at 37 °C for 30 min. After washing the sections with PBS, a SABC kit (SA1022, Boster Biological Technology, Wuhan, China) was added for 20 m at 37 °C. Sections were then dehydrated, cleared, and mounted after being washed with PBS, counterstained with hematoxylin, and mounted. Slides were dried in a fume hood overnight. Pictures were taken using a digital section scanner (KF-PRO-20 MAGSCANCNER KFBIO, KFBIO, Zhejiang, China).

## Immunofluorescence staining

The samples of mouse bladder cancer were first fixed in 4% paraformaldehyde, then permeabilized with 0.1% Triton X-100 (Sigma, USA), and then blocked with 1% bovine serum in PBS at room temperature. The sections were first stained with a primary antibody at a

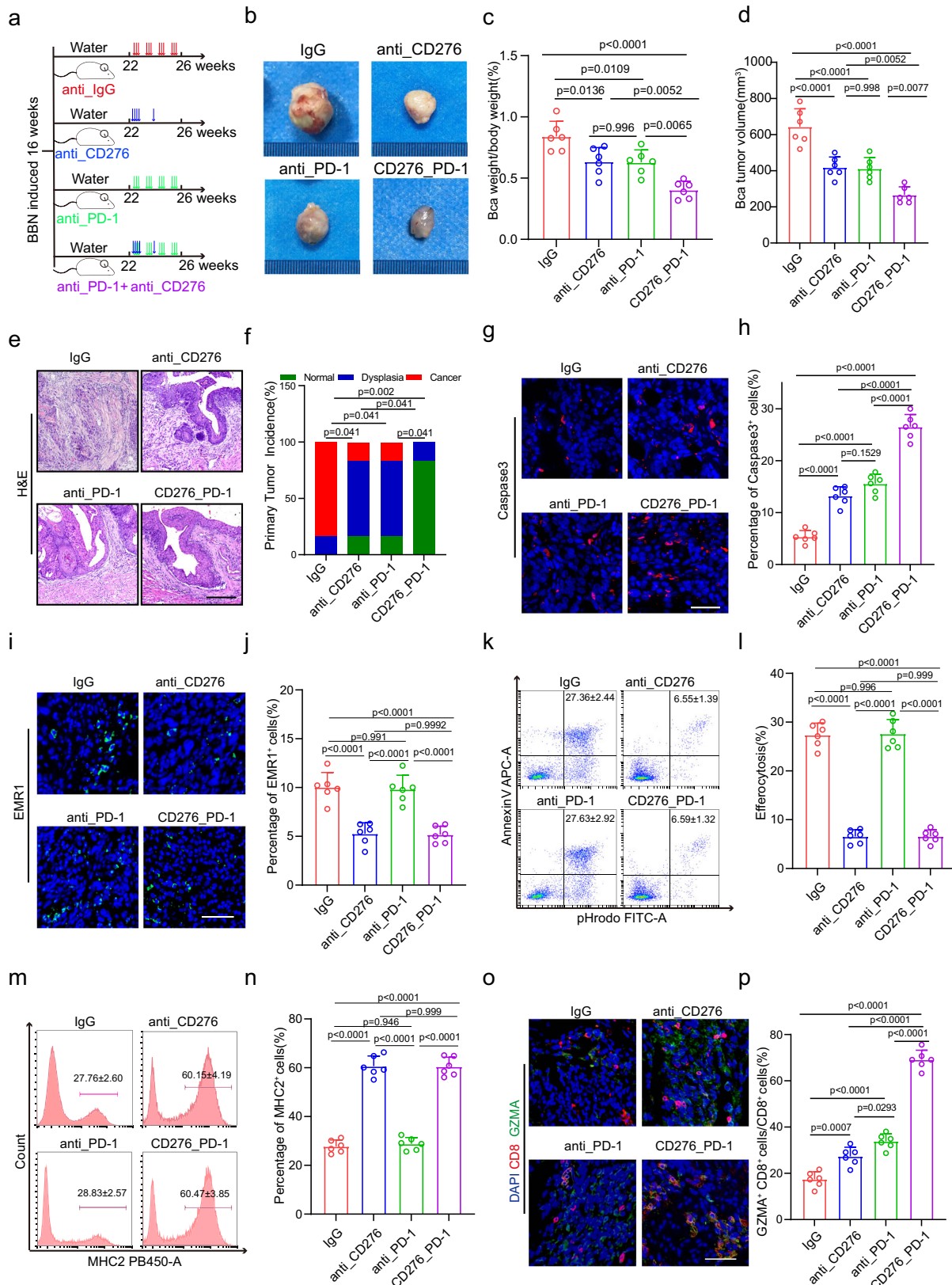

concentration of 1:100 and incubated at 4 °C overnight. Next, the corresponding secondary antibody conjugated with DyLight 488 and Fluor 594 was applied to the sections (Invitrogen, United States). To ensure the primary antibody's specificity, normal serum was also used as a control at the same time. 4′,6-diamidino-2- phenylindole (DAPI) was used as a counterstain for the nuclei at a ratio of 1:1000 for 1 min.

Pictures were taken using an upright fluorescent microscope (ZEISS LSM880 with Airyscan).

**Flow cytometry analysis and fluorescence-activated cell sorting**
As stated above, single cell suspensions of mouse marrow, spleen, and xenograft were created. Before staining, cells were resuspended at a

**Fig. 7 | CD276 enhances anti-PD-1-induced tumor regression. a** The experimental design of the bladder cancer tumorigenesis model and different strategies of treatment. The male mice were treated with either anti-CD276 antibody (10 mg/kg body weight, 3 times a week) with or without PD-1 antibody (200 μg/mouse, day1, 3, 5, 7, 14) as indicated starting from week 22. **b–d** Representative image of BBN-induced bladder carcinoma in male mice (**b**). Quantification of the ratio of BLCA weight to body weight (**c**) and BLCA tumor volume (**d**) in the different treatment groups. Data are presented as mean ± SD ($n = 6$). $P$ values were presented by one-way ANOVA with Tukey's multiple comparison test. Representative H&E staining of BLCA (**e**) and quantification of primary tumor incidence (**f**) in different treatment groups. Scale bar, 200 μm. $P$ values were calculated by Pearson chi-square test. **g–j** Representative Immunofluorescence (IF) staining images of Caspase3 (**g**) and

EMR1 (**i**). Quantification of percentages of Caspase3[+] (**h**) and EMR1[+] (**j**) cells in different treatment groups. Scale bar, 50 μm. Data was shown as mean ± SD ($n = 6$). $P$ values were presented by one-way ANOVA with Tukey's multiple comparison test. Representative flow cytometry plots (**k, m**) and quantification of the percentages of efferocytosis (**l**) and percentages of MHCII[+] cells (antigen presenting ability of macrophages, **n**) in macrophages of different treatment groups. Data was shown as mean ± SD ($n = 6$). $P$ values were presented by one-way ANOVA with Tukey's multiple comparison test. **o, p** Representative Immunofluorescence (IF) staining images of CD8 (red) and GZMA (green, **o**). Statistical analysis of the radios of CD8[+] GZMA[+] cells to CD8[+] cells (**p**) in different treatment groups. Scale bar, 50 μm. Data are shown as mean ± SD ($n = 6$). $P$ values were presented by one-way ANOVA with Tukey's multiple comparison test.

concentration of $1 \times 10^7$ cells/mL after being rinsed in staining buffer (2% bovine growth serum in PBS). Cells were stained with antibodies and then resuspended in staining buffer for 2 h at 4 °C to achieve extracellular staining. Anti-Mouse CD11C PE-CY7 (25-0114-81, ThermoFisher, 1:200); Anti-Mouse F4/80 PCP-CY5.5 (45-4801-80, Thermo Fisher, 1:200); Anti-Mouse Ly-6G (Gr-1) Alexa Fluor® 700 (56-5931-82, ThermoFisher, 1:200); Annexin V-APC/7-AAD Apoptosis kit (AP105, LIANKE, 1:200); Anti-Mouse MHC Class II (I-A/I-E) eFluor® 450 (48-5321-82, ThermoFisher, 1:200); Anti-Mouse CD45 PE-CY7(60-0451, Tonbo, 1:200); Anti-Mouse CD3 APC (20-0032, Tonbo, 1:200); Anti-Mouse CD4 violet Fluor™ 450 (75-0041, Tonbo, 1:200); Anti-Mouse CD8a FITC(35-0081, Tonbo, 1:200) and Ghost Dye™ Red 780 (13-0865, Tonbo, 1:100) were the antibodies and dyes used for flow cytometry (CytoFLEX, Beckman Coulter). Data were examined using NovoExpress software, and samples were examined using a flow cytometer (Cyto-FLEX, Beckman Coulter) (version 2.0). For FACS, single cell suspensions of digested tumor were Incubated for 15 m with 500 ng of Fc blocker (anti-CD16/32, E-AB-F0997A-100 μg, Elabscience, 1:100), and then for 30 m at 4 °C with Anti-Mouse CD11C PE-CYN7, Anti-Mouse F4/80 PCP-CYN5.5 and Anti-Mouse Ly-6G (Gr-1) Alexa Fluor® 700 (1:200). This procedure was done in order to sort intratumoral TAMs, MDSCs and DCs by flow cytometer (LSRFOrtessa, BD).

### Cell culture
MBT2 cells were obtained from the First Affiliated Hospital of Anhui Medical University, while MB49 cells were purchased from EMD Millipore (Merck, Cat# SCC148). The cells were cultured in DMEM (Gibco, USA) and RPMI 1640 (Gibco, USA), supplemented with 10% fetal bovine serum (FBS, Gibco, USA), 1% penicillin (Gibco, USA), and 1% streptomycin (Gibco, USA) at 37 °C in a water-saturated atmosphere under 5% $CO_2$ in an incubator (Thermo Scientific, USA).

### Efferocytosis assay
On the day of the assay, MB49 cells were irradiated with 254 nm ultraviolet lamp for 30 min, and then incubated with pHrodo (pHrodo™ Green E. coli BioParticles™, P35366, 2 mg/ml) at 37 °C 5% carbon dioxide for 2 h. This method usually produces 80−85% apoptotic cells. Apoptotic cells stained with Annexin V-APC/7-AAD Apoptosis kit (AP105) for 10 min and washed with PBS. Stained apoptotic cells were added into flow sorted cells (apoptotic cells: sorted cells) at a ratio of 4:1, incubated for 2 h at 37 °C with 5% $CO_2$, washed twice with PBS, and then scraped with cell scrapers. The efferocytosis capacity of sorted cells was analyzed by flow cytometry.

### Western blotting
The mouse bladder cancer tissues were dissociated into single cells, and TAMs were sorted by flow cytometry. TAMs were immediately lysed with ice-cold RIPA lysis buffer (P0013B, Beyotime, Jiangsu, China) with protease and phosphatase inhibitors (4693132001, Roche, Shanghai, China) using a gentleMACS dissociator in order to identify protein expression (130-093-235, Macs Miltenyi Biotec, Guangzhou, China). Using the BCA protein estimation technique, protein

concentration was calculated (23227, Thermo Fisher). Equivalent amounts of protein were electrophoretically separated on 10% SDS polyacrylamide gels (EpiZyme, PG11X) and then transferred to PVDF membranes (Merck Millipore, IPVH00010). AXL (AF7793, Affinity, 1:500), CD276 (A17216, Abclonal, 1:1000), MerTK (YT2733, Immuno-Way, 1:1000), GAPDH (2118 S, CST, 1:2000), LAPTM5 (YN4729, ImmunoWay, 1:1000), CLTC (26523-1-AP, Proteintech, 1:1000), LIPA (12956-1-AP, Proteintech, 1:1000), LAMP2 (27823-1-AP, Proteintech, 1:1000), CCR1 (PA1-41062, Thermo Fisher, 1:1000), CCR5 (YT6108, ImmunoWay, 1:1000), CX3CR1 (YT5112, ImmunoWay, 1:1000), CXCR4 (11073-2-AP, Proteintech, 1:1000), CXCL9 (22355-1-AP, Proteintech, 1:1000) and JUN (24909-1-AP, 1:2000) were probed on the blots, and after washing, they were again probed with a secondary anti-rabbit horseradish peroxidase antibody (SA00001-2, Proteintech, 1:2000). ECL was used to detect the bands (Tanon,180-5001).

### ChIP-qPCR
Cells were cross-linked with 1% paraformaldehyde for 10 min at room temperature to capture interacting proteins and DNA and then quenched with 0.125 M glycine. Cells are then lysed to release cellular components and to dissolve cross-linked protein-DNA complexes. Soluble chromatin was precipitated with anti-JUN (24909-1-AP, Proteintech, Wuhan, China) or a control IgG (B900610-1MG, Proteintech, Wuhan, China). For real-time qPCR was performed using PerfectStart® Green qPCR SuperMix (AQ601, TransGen Biotech), by a Bio-Rad CXF96 real-time system (Bio-Rad, USA). Data are reported as relative-fold enrichment. Primers used for ChIP qPCR are listed in Supplementary Data 6.

### Statistical analyses
In this study, numerical data and histograms were presented as the mean ± SD. Detailed statistical parameters of the analyses are stated in the Figure Legends. All experiments were performed at least twice. Data shown are either a single representative or combination from different independent experiments. Two-tailed Student's $t$ test was used between two groups and a difference was considered statistically significant with $p < 0.05$. For comparison multiple groups ($n > 2$), the differences were decided by one-way ANOVA. For survival analysis, the log-rank test was used for analysis. For incidence primary tumor analysis, $P$ value was calculated by Pearson chi-square test. All statistical analyses were conducted using GraphPad Prism 9 and SPSS software.

### Reporting summary
Further information on research design is available in the Nature Portfolio Reporting Summary linked to this article.

## Data availability
The scRNA-seq data generated in this study have been deposited in the Genome Sequence Archive in the National Genomics Data Center under the accession number CRA008674. The previously published scRNA-seq data of GEO, GSE135337 were used in this paper. The remaining data are available within the Article, Supplementary

Information or Source Data file. Further information and requests for resources and reagents should be directed to and will be fulfilled by the Lead Contact, Liang Peng (pengliang@301hospital.com.cn). Source data are provided with this paper.

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

## Acknowledgements

We thank Dr. Guoqiang Yuan, Dr. Xiaojun Xia and Dr. Cheng Wang for provide mice, reagents and plasmids. This work is supported by National Key R&D Program of China (2022YFA1105300, 2021YFC2501000), Grant 21-163-12-ZT-006-002-05, Grant 2021-JCJQ-ZD-077-11, Guangzhou Municipal Science and Technology Bureau (2024B03J1384), National Natural Science Foundation of China (82173362, 81872409, U22A20331, 82273414 and 81974443).

## Author contributions

Conceptualization, M.C. S.C., D.C., Y.L. and L.P.; Methodology, M.C., S.C., K.L., G.W., G.X., R.L., C.Z., Z.Z., Z.C., X.W., Y.L. and S.L.; Data Analysis and Curation, M.C., S.C., K.L., G.W., G.X., R.L., C.Z., Z.Z., H.H., G.T., Z.C., X.W., Y.L., R.Z., H.X., Y.L. and D.C.; Investigation and Validation, M.C., K.L., G.W., H.X., Y.L., J.M. and D.C.; Resources, Y.Z., J.L., D.C., Y.L. and L.P.; Writing, M.C. S.C., D.C., Y.L., J.M. and L.P. Supervision and funding acquisition, S.L., Y.L., D.C. and L.P.

## Competing interests

The authors declare no competing interests.
