## [Peer Review File · Nature Communications]

CD276-dependent efferocytosis by tumor-associated macrophages promotes immune evasion in bladder cancerREVIEWER COMMENTS

Reviewer #1 (Remarks to the Author):

In this article NCOMM-23-05637 "CD276 expression by tumor associated macrophages promotes efferocytosis to confer immune evasion in cancer", Chen and colleagues report on a biochemical, cells biological, and in vivo tumorigenesis/immunotherapy study to investigate global and conditional loss of function of CD276, a member of the B7 superfamily of receptors associated with cancer progression and metastasis in multiple tumor types, in a murine bladder cancer model. Authors first generate a whole-body CD276 KO to investigate tumorigenesis and OS in a chemical carcinogenesis BBN bladder carcinoma model, and then interrogate the tumor microenvironment by immune filtration and scRNAseq. Main conclusions in the paper include findings that whole-body CD276 KO significantly inhibits tumorigenesis and promotes T cell infiltration, and this expression pattern appears to be phenocopied in human cancer assessed by TCGA data analysis. Further single cell transcriptomics and pathway analysis indicate the CD276 is expressed on both tumor cells (epithelial cells) and macrophages, and conditional KO of CD276 in macrophages alters infiltration of macrophages, increases MHC II expression, and generally phenocopies effects of whole-body KO. In second half of the paper, authors again use scRNA transcriptomics and CellChat analysis to assess mechanistic insight and propose a macrophage axis linking CD276 with a Jun->Gas6->Mertk->Axl that drives efferocytosis and immune escape. Finally, conditional KI of CD276 augments this pathway, pan-Axl/Merk (R428) inhibitors reverse this pathway, and anti-CD276 and antiPD1 synergize.

Overall, this is comprehensive and robust study, that covers a lot of ground. The knockout approaches and knockin approaches to manipulate CD276 are technically well performed and support of robust cancer and IO phenotype in the BBN model. The association of CD276 with tumor associated macrophages is also very interesting and further supported by the cKO model. The second part of the paper is less convincing and the link between CD276 KO and Jun/Gas6/Mertk/Axl and efferocytosis is premature from what is presented. The relationship between CD276 expression on tumors (intrinsic) and macrophage (extrinsic) also needs better explanation (for example, what is the phenotype when CRISPR KO MB49 cells are xenografted into WT or CD276 KO mice). Collectively, while a role of CD276 in IO is supported, the efferocytosis connection is still preliminary from what is presented. If a better connection between CD276 loss of function and Mertk/Axl could be presented with additional data, the paper would be better suited for NCOMM.

Reviewer #2 (Remarks to the Author):

In this manuscript titled "CD276 expression by tumor-associated macrophages promotes efferocytosis to confer immune evasion in cancer", Maosheng Cheng and co-authors show that both mouse and human TAMs in bladder cancer express high levels of CD276. CD276 expression is associated with the poor survival of cancer patients. To evaluate the role of CD276 in the progression of bladder cancer, authors utilized a variety of research methods, including immunohistochemistry, flow cytometry, fluorescent microscopy, single-cell RNAseq, ChIP-qPCR, CD276 knockout mice, BBN-induced carcinogenesis model, etc. Authors demonstrate that CD276-mediated signaling activates JUN to promote the expression of AXL/MERTK in TAMs, which in turn induces the efferocytosis of TAMs and inhibits the MHC II-mediated antigen-presenting of TAMs. The authors propose that targeting CD276 could serve as an alternative approach to bladder cancer therapy. However, this manuscript has several concerns that need to be addressed.

1. Fig. 7A shows that injections of anti-PD1 and anti-CD276 Abs in mice were done between 22 and 26 wk after initiation of BBN-induced carcinogenesis. Legend for Fig. 7 is missing specific information on an antibody treatment schedule, however in section Methods (lines 480-481) authors indicate that administration of anti-PD1 abs was done on Days 1, 3, 5, 7, 14???. The authors should explain this evident discrepancy in the protocols.

2. It's not clear how many mice were tumor-free after treatment with anti-PD1 and CD276 Abs. Authors claim that such a combination has a significant therapeutic potential for bladder cancer. To prove it fully, authors should provide additional evidence of the therapeutic efficacy of such treatment using syngeneic murine models of bladder cancer (MB49 for C57/Bl6 mice and MBT2 for C3/H mice). These experiments are fairly quick and can be done within 1-2 months. The therapeutic experiments should be done using tumor-bearing mice with established tumors. And survival curve will provide important additional information on the therapeutic efficacy of the proposed treatment. The obtained data also will confirm whether CD276 has "unprecedented" roles (line 39) in bladder cancer.

3. The authors demonstrate that expression of CD276 was found in all cell types in the mouse model of bladder cancer (lines 186-187). If so, why the manuscript is focused on TAMs expressing CD276?

4. Typically, the most common marker for tumor-associated macrophages in mice is F4/80, not CD68. Do the F4/80+ TAMs co-express CD276?

5. The manuscript should be carefully edited to correct multiple errors and typos.

Reviewer #3 (Remarks to the Author):

In this manuscript, the authors focused on unveiling the role of CD276, one of the immune checkpoint, on tumor-associated macrophages and thereby in anti-tumor immune response. The authors claim that genetic ablation of CD276 on TAM is accompanied by reduced efferocytosis, enhanced MHC II expression and increased CD8+ T cell infiltration. Furthermore authors claim that blockade of CD276 along with PD-1 can synergistically restrain tumor growth. The findings are interesting. However, there are experimental inconsistencies, lack of mechanistic insights and the use of single mouse model used are not sufficiently specific to draw definitive conclusions.

Main points:

1. In the manuscript (Fig. 1), the authors show that high expression of CD276 co-relates with poor clinical outcome in BLCA. The authors further show that ablation of blockade of CD276 results in reduced tumor burden and improved survival. Is this a pan-cancer discovery conserved in other tumor models, like pancreatic, colon, wherein high expression of CD276 is also co-related with poor survival?

2. In Figure 2 and Figure 3, wKO as well as cKO the proportion of infiltrated TAM is reduced compared to corresponding controls. Using cKO, it can be inferred that CD276 on myeloid cells can play a role in their infiltration. However, using specific KO (eg. K5), role of CD276 expressed on other cell in macrophage recruitment should be ruled out.

3. Can authors provide mechanistic insight for reduced TAM infiltration (CCL2, CX3CL1, PAI-1, etc)

4. Since Lys-M as well as well CD276 is expressed on myeloid cells, it should be confirmed if the CD276 KO is specifically on TAM and not on other tumor-associated myeloid cells such as MDSC and the current functional effect is also TAM-specific?

5. The authors show that myeloid (TAM)-specific CD276 lead to enhanced MHC II expression and enhanced CD8+ T cell infiltration and activation. How about CD4+ T cells? Additionally such TAM with high expression MHC II should be sorted and should be directly analysed for their ability to prime/activate CD4/CD8 T cells. Can authors provide mechanistic insight for enhanced CD8+ T cell infiltration (CXCL9, etc)

6. For efferocytosis, a more sophisticated tools such as pHrodo dye should be used to provide clear evidence. Nonetheless, for efferocytosis assay a detailed gating strategy should be provided, to confirm only "efferocytosed" Annexin V+ cells are analysed.

7. The authors have used CD68 as a marker for sorting TAM. CD68 is primarily expressed intracellularly, with only fewer percentage expressing CD68 at surface. Furthermore, since CD68 is also expressed by other cells of monocytic lineage, the detailed gating strategy reported in the literature should be considered for TAM isolation and/or analysis.

Minor Points:

1. Proof reading of manuscript is required. There are some grammatical mistakes as well as inconsistencies with figure number in the text (e.g. Fig. S4C and S4D on line 252-255).
2. Tumor volume (at least at sacrifice) for Fig. 1H, 3C, 6C, 7B etc.)
3. Bar diagram for Fig. 2H
4. Gating strategy for sorting of cells as well as for other flow cytometry analysis. Also please refer to comment above regarding markers used.
5. Survival analysis (Kaplan Meier curve) for cKI, R428 as well CD276+a-PD-1 analysis

We would like to express our sincere gratitude to the reviewers for their invaluable comments and insights. We have wholeheartedly taken their critiques and suggestions into account and have diligently addressed their concerns. Substantial additions of new data have been incorporated into both primary and supplemental figures, effectively addressing the raised points from the initial review. The manuscript has greatly benefited from these efforts, resulting in a stronger and clearer presentation of our novel findings. We extend our gratitude to the reviewers for their assistance in further strengthening our research. Additionally, we appreciate the reviewers' patience as the revision process required more time than anticipated due to the extensive experimental work involved. We are thankful for their understanding and willingness to consider the revised manuscript.

REVIEWER COMMENTS

Reviewer #1 (Remarks to the Author):

In this article NCOMM-23-05637 “CD276 expression by tumor associated macrophages promotes efferocytosis to confer immune evasion in cancer”, Chen and colleagues report on a biochemical, cells biological, and in vivo tumorigenesis/immunotherapy study to investigate global and conditional loss of function of CD276, a member of the B7 superfamily of receptors associated with cancer progression and metastasis in multiple tumor types, in a murine bladder cancer model. Authors first generate a whole-body CD276 KO to investigate tumorigenesis and OS in a chemical carcinogenesis BBN bladder carcinoma model, and then interrogate the tumor microenvironment by immune filtration and scRNAseq. Main conclusions in the paper include findings that whole-body CD276 KO significantly inhibits tumorigenesis and promotes T cell infiltration, and this expression pattern appears to be phenocopied in human cancer assessed by TCGA data analysis. Further single cell transcriptomics and pathway analysis indicate the CD276 is expressed on both tumor cells (epithelial cells) and macrophages, and conditional KO of CD276 in macrophages alters infiltration of macrophages, increases MHC II expression, and generally phenocopies effects of whole-body KO. In second half of the paper, authors again use scRNA transcriptomics and CellChat analysis to assess mechanistic insight and propose a macrophage axis linking CD276 with a Jun->Gas6->Mertk->Axl that drives efferocytosis and immune escape. Finally, conditional KI of CD276 augments this pathway, pan-Axl/Merk (R428) inhibitors reverse this pathway, and anti-CD276 and antiPD1 synergize.

Overall, this is comprehensive and robust study, that covers a lot of ground. The knockout approaches and knockin approaches to manipulate CD276 are technically well performed and support of robust cancer and IO phenotype in the BBN model. The association of CD276 with tumor associated macrophages is also very interesting and further supported by the cKO model. The second part of the paper is less

convincing and the link between CD276 KO and Jun/Gas6/Mertk/Axl and efferocytosis is premature from what is presented. The relationship between CD276 expression on tumors (intrinsic) and macrophage (extrinsic) also needs better explanation (for example, what is the phenotype when CRISPR KO MB49 cells are xenografted into WT or CD276 KO mice). Collectively, while a role of CD276 in IO is supported, the efferocytosis connection is still preliminary from what is presented. If a better connection between CD276 loss of function and Mertk/Axl could be presented with additional data, the paper would be better suited for NCOMM.

Response: Thank you sincerely for your comprehensive and encouraging assessment of our study. We greatly appreciate your recognition of the technical robustness and the extensive ground covered in our investigation of CD276. Regarding the second part of the paper, we take your evaluation very seriously. We acknowledge that the link between CD276 knockout and Jun/Gas6/Mertk/Axl, and efferocytosis, may not have been convincingly established based on the current presentation. In response to your valuable feedback, we have undertaken additional experiments. Specifically, we transplanted CRISPR KO MB49 cells into both WT and CD276 KO mice, and interestingly, IHC staining results showed that CD276 deficiency increased the apoptosis level of MB49 and did not affect the proliferation of MB49 in CD276 wKO mice. More importantly, compared to MB49-SG, we found that CD276 deficiency inhibited the infiltration of EMR1⁺ macrophages in CD276 wKO mice (Fig. R1).

Fig. R1. **a** Representative images of the staining of H&E, KI67, Caspase3, EMR1 in the different groups. Scale bar, 100 μ m. **b** Quantification of the percentage of KI67⁺, Caspase3⁺ and EMR1⁺ cells in the different groups. Data are presented as mean \pm SD (n=6). *P* values are presented by one-way ANOVA with Tukey's multiple comparison test.

Furthermore, to bolster the connection between CD276 and Mertk/Axl, we performed pathway analysis and carefully examined the impact of CD276 deletion on key proteins within the lysosomal pathway. Through Western blot validation, our results showed that CD276 deletion reduced the protein expression levels of CLTC (Clathrin Heavy Chain), LIPA (Lipase A, Lysosomal Acid Type), LAMP5 (Lysosomal Protein Transmembrane 5) and LAMP2 (Lysosomal Associated Membrane Protein 2), which are key regulators of the lysosomal and phagocytic signaling pathways (New Fig. 4g), which further support the association between CD276 and efferocytosis. Our results further confirm that CD276 deletion reduces protein expression of chemokines including CCR1 (C-C Motif Chemokine Receptor 1), CCR5 (C-C Motif Chemokine Receptor 5), CX3CR1 (C-X3-C Motif Chemokine Receptor 1) and CXCR4 (C-X-C Motif Chemokine Receptor 4), which have been shown to promote macrophage infiltration¹⁻⁴ (New Fig. 4h).

New Fig. 4. **g** Western blot of CLTC, LIPA, LAMP5 and LAMP2 in control and cKO groups using GAPDH as loading control. **h** Western blotting for CCR1, CCR5, CX3CR1 and CXCR4 in control and cKO groups with GAPDH as loading control.

We genuinely appreciate your encouragement and guidance. Your invaluable suggestions have prompted us to conduct further experiments and analyses to enhance the quality of our research. We are confident that with these improvements, our study will be better suited for publication in NCOMM.

Reviewer #2 (Remarks to the Author):

In this manuscript titled "CD276 expression by tumor-associated macrophages promotes efferocytosis to confer immune evasion in cancer", Maosheng Cheng and co-authors show that both mouse and human TAMs in bladder cancer express high levels of CD276. CD276 expression is associated with the poor survival of cancer patients. To evaluate the role of CD276 in the progression of bladder cancer, authors utilized a variety of research methods, including immunohistochemistry, flow cytometry, fluorescent microscopy, single-cell RNAseq, ChIP-qPCR, CD276 knockout mice, BBN-induced carcinogenesis model, etc.

Authors demonstrate that CD276-mediated signaling activates JUN to promote the expression of AXL/MERTK in TAMs, which in turn induces the efferocytosis of TAMs and inhibits the MHC II-mediated antigen-presenting of TAMs. The authors propose that targeting CD276 could serve as an alternative approach to bladder cancer therapy.

However, this manuscript has several concerns that need to be addressed.

Response: We sincerely appreciate your careful evaluation of our manuscript and the positive aspects you highlighted regarding our study on CD276 in bladder cancer and its potential implications for therapy. We have carefully considered your concerns, and we would like to address them in our response. We believe the new version provides more comprehensive evidence supporting the role of CD276 in bladder cancer and its therapeutic implications.

1. Fig. 7A shows that injections of anti-PD1 and anti-CD276 Abs in mice were done between 22 and 26 wk after initiation of BBN-induced carcinogenesis. Legend for Fig. 7 is missing specific information on an antibody treatment schedule, however in section Methods (lines 480-481) authors indicate that administration of anti-PD1 abs was done on Days 1, 3, 5, 7, 14???. The authors should explain this evident discrepancy in the protocols.

Response: Thank you for pointing out the discrepancy in the antibody treatment schedule in Fig. 7a and the missing specific information in the figure legend. We have corrected this in the updated manuscript.

2. It's not clear how many mice were tumor-free after treatment with anti-PD1 and CD276 Abs. Authors claim that such a combination has a significant therapeutic potential for bladder cancer. To prove it fully, authors should provide additional evidence of the therapeutic efficacy of such treatment using syngeneic murine models of bladder cancer (MB49 for C57/B16 mice and MBT2 for C3/H mice). These experiments are fairly quick and can be done within 1-2 months. The therapeutic experiments should be done using tumor-bearing mice with established tumors. And

survival curve will provide important additional information on the therapeutic efficacy of the proposed treatment. The obtained data also will confirm whether CD276 has “unprecedented” roles (line 39) in bladder cancer.

Response: Thank you for your valuable suggestions. In response, we have taken your advice seriously and conducted therapeutic experiments using MBT2 cells for C3/H mice and MB49 cells for C57/B16 mice, which further confirmed the potential of CD276 in bladder cancer therapy. Additionally, we have included survival curve data, which provides essential information on the therapeutic efficacy of the proposed treatment (New Supplementary Fig. 7b, c).

New Supplementary Fig. 7. **b** Kaplan-Meier curve of overall survival in different treatment groups of C57 tumour-bearing mice injected with MB49 cells. **c** Kaplan-Meier curve of overall survival in different treatment groups of CH3 tumour-bearing mice injected with MBT2 cells. P value was calculated by log-rank test.

3. The authors demonstrate that expression of CD276 was found in all cell types in the mouse model of bladder cancer (lines 186-187). If so, why the manuscript is focused on TAMs expressing CD276?

Response: Thank you for raising the question regarding the focus on tumor-associated macrophages (TAMs) expressing CD276 in our manuscript. We appreciate your insightful inquiry and would like to address it with relevant data. While it is true that CD276 expression was found in all cell types in the mouse model of bladder cancer, our research has revealed that the impact of CD276 knockout in the tumor microenvironment is more significant in TAMs compared to tumor cells (Supplementary Fig. 2 m-o). Through comprehensive studies, including whole-body CD276 knockout, we observed that the number of TAMs in the tumor microenvironment underwent the most significant changes following CD276 knockout (Fig. 2d). This finding suggests that CD276 plays a crucial role in regulating TAMs and their functions in the tumor microenvironment. Additionally, we conducted supplementary experiments involving conditional knockout of CD276 in fibroblasts and endothelial cells. Interestingly, we found that CD276 knockout in these cell types did not affect tumor formation (Fig. R2). This result further supports our focus on

TAMs expressing CD276 as they appear to have a more pronounced impact on tumor development and immune evasion. Based on the data obtained from these experiments, we believe that the concentration on CD276 expression in TAMs is justified as it demonstrates the pivotal role of CD276 in shaping the tumor microenvironment and promoting immune evasion in cancer.

Supplementary Fig. 2. **m-o** Representative images of tumors (**m**), tumor growth curves (**n**) and tumor weight (**o**) at the sacrifice of subcutaneous xenograft model.

Fig. 2d Bar plots of proportion of cell type (left) and total cell number (right) in WT or wKO group.

Fig. 2. **d** Bar plots of proportion of cell type (left) and total cell number (right) in WT or wKO group.

Fig. R2. **a, b** Representative images of tumors in a xenograft mouse model with subcutaneous implantation of MB49 cells from WT and cKO (Col1a2creERT/Cdh5creERT) mice. Tumor growth curves and tumor weight at the sacrifice of subcutaneous xenograft model.

4. Typically, the most common marker for tumor-associated macrophages in mice is F4/80, not CD68. Do the F4/80+ TAMs co-express CD276?

Response: Thank you for thorough review to our research. In our revised manuscript, we have adopted (EMR1) F4/80 as the marker for tumor-associated macrophages.

Through Opal/TSA multicolor IF analysis, we have investigated whether EMR1+ TAMs co-express CD276. The experimental results indicate that EMR1+ TAMs co-express CD276 in the tumor microenvironment and while CD276 deletion reduces the infiltration of EMR1+ TAMs. (New Fig. 2g). This finding further strengthens our understanding of the pivotal role of CD276 in tumor immune evasion.

New Fig. 2. **g** Opal/TSA multicolor IF staining with anti-CD276 and EMR1 antibodies (left). Nuclei are stained with DAPI (blue) and quantification of percentages of EMR1+ cells (right) in WT and wKO mice. Scale bar, 100 μ m.

5. The manuscript should be carefully edited to correct multiple errors and typos.

Response: Thank you for your evaluation and valuable feedback on our manuscript. In response to your comments, we have taken your feedback seriously and conducted a thorough review and editing process to address these issues. We have carefully corrected the errors and typos throughout the manuscript to ensure clarity and accuracy in our presentation.

Reviewer #3 (Remarks to the Author):

In this manuscript, the authors focused on unveiling the role of CD276, one of the immune checkpoint, on tumor-associated macrophages and thereby in anti-tumor immune response. The authors claim that genetic ablation of CD276 on TAM is accompanied by reduced efferocytosis, enhanced MHC II expression and increased CD8+ T cell infiltration. Furthermore authors claim that blockade of CD276 along with PD-1 can synergistically restrain tumor growth. The findings are interesting. However, there are experimental inconsistencies, lack of mechanistic insights and the use of single mouse model used are not sufficiently specific to draw definitive conclusions.

Response: Thank you very much for your appreciation and valuable feedback on our manuscript. We truly value your interest in our research regarding the role of CD276, an immune checkpoint, in tumor-associated macrophages (TAMs) and its impact on anti-tumor immune responses. Your positive evaluation means a lot to us, and we sincerely appreciate it. We also take your concerns about experimental inconsistencies, lack of mechanistic insights, and the use of a single mouse model very seriously. In response to your feedback, we have thoroughly addressed and supplemented these aspects in the revised manuscript.

Main points:

1. In the manuscript (Fig. 1), the authors show that high expression of CD276 co-relates with poor clinical outcome in BLCA. The authors further show that ablation of blockade of CD276 results in reduced tumor burden and improved survival. Is this a pan-cancer discovery conserved in other tumor models, like pancreatic, colon, wherein high expression of CD276 is also co-related with poor survival?

Response: Thank you for your evaluation and valuable suggestion. You raised an important question regarding the correlation of high CD276 expression with poor clinical outcomes in other tumor models, such as pancreatic and colon cancer. In response, we conducted an analysis using TCGA pan-cancer data to explore the expression and survival association of CD276 in other tumor types. Our analysis shows that CD276 is highly expressed in tumor tissues of various tumor types compared to adjacent tissues. Moreover, the high expression of CD276 is closely related to the poor clinical outcome of many tumor types, including colorectal cancer, but not pancreatic cancer (Figure R3). This finding reinforces our preliminary observation of the correlation between CD276 and clinical cancer prognosis.

Fig R3. **a, b** CD276 gene expression in pan-cancer (a). Survival analysis of CD276 in pan-cancer (b). Show only tumor types that are statistically significant.

2. In Figure 2 and Figure 3, wKO as well as cKO the proportion of infiltrated TAM is reduced compared to corresponding controls. Using cKO, it can be inferred that CD276 on myeloid cells can play a role in their infiltration. However, using specific KO (eg. K5), role of CD276 expressed on other cell in macrophage recruitment should be ruled out.

Response: Thank you for your insightful feedback. In response to your concern, we have conducted additional experiments to address this issue. We have now provided data showing that after CD276 knockout in the tumor microenvironment are more significant compared to the changes in tumor cells alone (Supplementary Fig. 2 m-o). Furthermore, we observed that the reduction in the number of macrophages was most pronounced in the wKO model (Fig. 2d), indicating the crucial role of macrophages in the tumor microenvironment. Additionally, we have included experiments involving conditional knockout of CD276 in fibroblasts and endothelial cells (Fig. R2). The results demonstrated that CD276 knockout in these cell types did not affect tumor formation, which supports our conclusion that other cell types expressing CD276 do not play a significant role in macrophage recruitment.

Supplementary Fig. 2 **m-o** Representative images of tumors (m), tumor growth curves (n) and tumor weight (o) at the sacrifice of subcutaneous xenograft model.

Fig. 2 **d** Bar plots of proportion of cell type (left) and total cell number (right) in WT or wKO group.

Fig. R2 **a, b** Representative images of tumors in a xenograft mouse model with subcutaneous implantation of MB49 cells from WT and cKO (*Col1a2creERT/Cdh5creERT*) mice. Tumor growth curves and tumor weight at the sacrifice of subcutaneous xenograft model.

3. Can authors provide mechanistic insight for reduced TAM infiltration (CCL2, CX3CL1, PAI-1, etc)

Response: Thank you for your valuable suggestion. In response, we conducted a pathway analysis to investigate the relevant mechanistic pathways and subsequently validated them through western blot experiments. Our results showed strong downregulation of chemokine signaling pathways in CD276 cKO TAMs (New Fig. 4f). To support these predictions, we performed western blot experiments. Our results further confirm that CD276 deletion reduces protein expression of chemokines including CCR1, CCR5, CX3CR1 and CXCR4, which have been shown to promote macrophage infiltration¹⁻⁴ (New Fig. 4h).

New Fig. 4. **f, h** KEGG pathway enrichment analysis using the intergroup differential gene of macrophages (f). Western blotting for CCR1, CCR5, CX3CR1 and CXCR4 in control and cKO groups with GAPDH as loading control (h).

4. Since Lys-M as well as well CD276 is expressed on myeloid cells, it should be confirmed if the CD276 KO is specifically on TAM and not on other tumor-associated myeloid cells such as MDSC and the current functional effect is also TAM-specific?

Response: Thank you for your insightful comments. In response to your query, we conducted additional experiments to address this issue. We carefully sorted and separated MDSCs, macrophages, and DC cells, and found that only macrophages showed significant differences in response to CD276 knockout (New Supplementary Fig. 4a-c). In addition, the antigen-presenting ability and efferocytosis function of macrophages showed significant changes. (New Fig. 4i, New Supplementary Fig. 4i, j, New Fig. 5i-k). These new findings confirm the specificity of CD276 knockout on TAMs and demonstrate that the observed functional effects are indeed TAM-specific.

New Supplementary Fig. 4. **a-c** Representative flow cytometry plots and statistical analysis of DCs (a), MDSCs (b) and macrophages (c) in control and cKO groups.

New Fig. 4. **i** Representative flow cytometry plots (left) and statistical analysis of MHCII⁺ macrophages (right) in control and cKO groups.

New Supplementary Fig. 4. **i, j** Representative flow cytometry plots and statistical analysis of MHCII⁺ cells in MDSCs (i) and DCs (j) from control and cKO groups.

New Fig. 5. **i-k** Representative flow cytometry plots and quantification of percentage efferocytosis in DCs (i), MDSCs (j) and macrophages (k) from control and cKO mice.

5. The authors show that myeloid (TAM)-specific CD276 lead to enhanced MHC II

expression and enhanced CD8⁺ T cell infiltration and activation. How about CD4⁺ T cells? Additionally such TAM with high expression MHC II should be sorted and should be directly analysed for their ability to prime/activate CD4/CD8 T cells. Can authors provide mechanistic insight for enhanced CD8⁺ T cell infiltration (CXCL9, etc)

Response: Thank you for your insightful feedback. In response to your suggestions, we have conducted flow cytometry to investigate the infiltration of CD4⁺ and CD8⁺ T cells. The results indicate that CD276 in myeloid cells indeed affects the infiltration of CD4⁺ and CD8⁺ T cells (New Fig. 4k), and we have included these findings in the revised manuscript.

New Fig. 4. k Representative flow cytometry plots (left) and statistical analysis of CD3⁺CD4⁺ T cells and CD3⁺CD8⁺ cells (right) in control and cKO groups.

To further analyze the ability of MHC II⁺ TAMs to prime/activate CD4/CD8 T cells and gain mechanistic insights into the enhanced CD8⁺ T cell infiltration, We collected MHCII⁺ TAMs and MHCII⁻ TAMs using flow cytometry. Subsequently, We observed a increase in protein levels of chemokine CXCL9 (C-X-C Motif Chemokine Ligand 9) in MHCII⁺ TAMs, which has been confirmed to be related to T cell recruitment⁵ (New Fig. 4j).

New Fig. 4. j Western blot of CXCL9 in MHCII⁺ TAMs and MHCII⁻ TAMs using GAPDH as loading control.

6. For efferocytosis, a more sophisticated tools such as pHrodo dye should be used to provide clear evidence. Nonetheless, for efferocytosis assay a detailed gating strategy should be provided, to confirm only “efferocytosed” Annexin V⁺ cells are analysed.

Response: Thank you for your constructive comments. We have redesigned the experiment using pHrodo dye to further validate our efferocytosis assays, and the results have been included in the revised manuscript (New Fig. 5h)

New Fig. 5 h Flow chart of efferocytosis. The apoptotic cells induced by irradiation were co cultured with the sorted cells for 2h and then analyzed by flow cytometry.

Additionally, we have provided a detailed gating strategy to confirm that only "efferocytosed" Annexin V⁺ cells are analyzed. This ensures the accuracy and specificity of our efferocytosis data. (New Supplementary Fig. 7h)

New Supplementary Fig. 7. h Sequential gating strategy of myeloid cells from tumor tissue for flow cytometry analysis. The respective gate names are given in the corresponding figures (h).

7. The authors have used CD68 as a marker for sorting TAM. CD68 is primarily expressed intracellularly, with only fewer percentage expressing CD68 at surface. Furthermore, since CD68 is also expressed by other cells of monocytic lineage, the

detailed gating strategy reported in the literature should be considered for TAM isolation and/or analysis.

Response: Thank you for your professional guidance. We understand that CD68 is primarily expressed intracellularly, with only a small percentage expressing CD68 on the cell surface. Additionally, since CD68 is also expressed by other cells of the monocytic lineage, it is important to consider using the detailed gating strategy reported in the literature for TAM isolation and/or analysis. In our revised manuscript, we have completely replaced CD68 with F4/80 as the specific marker for sorting TAMs. This ensures that we obtain more accurate and reliable results in TAM isolation, avoiding any potential shortcomings associated with CD68.

Minor Points:

1. Proof reading of manuscript is required. There are some grammatical mistakes as well as inconsistencies with figure number in the text (e.g. Fig. S4C and S4D on line 252-255).

Response: Thank you for your valuable comments. We have carefully revised the manuscript to correct any grammatical errors and ensure the consistency of figure numbers throughout the text. We have also cross-checked all figure references to ensure accuracy and clarity.

2. Tumor volume (at least at sacrifice) for Fig. 1H, 3C, 6C, 7B etc.)

Response: Thank you for your valuable comments. We have now included the tumor volume data in the revised manuscript for the mentioned figures.

3. Bar diagram for Fig. 2H

Response: Thank you for your valuable comments. We have now merged the original fig2g and h into a new fig2g in the revised draft and added a bar chart. This addition provides a clear and visual representation of the data and strengthens the interpretation of the results.(New Fig. 2g)

Fig. 2. **g** Opal/TSA multicolor IF staining with anti-CD276 and EMR1 antibodies (left). Nuclei are stained with DAPI (blue) and quantification of percentages of EMR1⁺ cells (right) in WT and wKO mice. Scale bar, 100 μ m.

4. Gating strategy for sorting of cells as well as for other flow cytometry analysis. Also please refer to comment above regarding markers used.

Response: Thank you for your valuable comments. We understand the importance of providing a clear gating strategy for sorting cells and other flow cytometry analysis, as well as addressing the previous comment regarding the markers used. We have now included a detailed gating strategy in the Supplementary figure for all flow cytometry analyses, including cell sorting experiments (Supplementary Fig. 7h, i). This will ensure transparency and reproducibility in our data analysis. Furthermore, we have taken into consideration the comment regarding the markers used. We have replaced CD68 with F4/80 as the specific marker for sorting TAMs, ensuring greater accuracy and specificity in TAM isolation.

Supplementary Fig. 7. **h, i** Sequential gating strategy of myeloid cells from tumour tissue for flow cytometry analysis. The respective gate names are given in the corresponding figures (h). Sequential gating strategy of lymphocytes from tumour tissue for flow cytometry analysis (i).

5. Survival analysis (Kaplan Meier curve) for cKI, R428 as well CD276+a-PD-1 analysis

Response: Thank you for your valuable comments. We have conducted the survival

analysis and generated Kaplan-Meier curves for these specific experimental conditions. The survival data for cKI, R428, and CD276+a-PD-1 analysis have been included in the revised manuscript (New Fig. 6c, New Supplementary Fig. 7a).

New Fig. 6. **c** The Kaplan-Meier overall survival curve in different treatment groups.

New Supplementary Fig. 8. **a** The Kaplan-Meier curve of the overall survival rate in the different treatment groups of BBN-induced BLCA.

REFERENCES

- 1 Tapmeier, T. T. *et al.* Evolving polarisation of infiltrating and alveolar macrophages in the lung during metastatic progression of melanoma suggests CCR1 as a therapeutic target. *Oncogene* **41**, 5032-5045, doi:10.1038/s41388-022-02488-3 (2022).
- 2 Chen, Z. *et al.* CCR5 signaling promotes lipopolysaccharide-induced macrophage recruitment and alveolar developmental arrest. *Cell Death Dis* **12**, 184, doi:10.1038/s41419-021-03464-7 (2021).
- 3 Busada, J. T. *et al.* Endogenous glucocorticoids prevent gastric metaplasia by suppressing spontaneous inflammation. *J Clin Invest* **129**, 1345-1358, doi:10.1172/JCI123233 (2019).
- 4 Werner, Y. *et al.* Cxcr4 distinguishes HSC-derived monocytes from microglia and reveals monocyte immune responses to experimental stroke. *Nat Neurosci* **23**, 351-362, doi:10.1038/s41593-020-0585-y (2020).
- 5 Dangaj, D. *et al.* Cooperation between Constitutive and Inducible Chemokines Enables T Cell Engraftment and Immune Attack in Solid Tumors. *Cancer Cell* **35**, 885-900 e810, doi:10.1016/j.ccell.2019.05.004 (2019).

REVIEWER COMMENTS

Reviewer #1 (Remarks to the Author):

This is a revised MS investigating role of B7 superfamily receptor CD276 in IO. To their credit, authors have successfully completely an important control investigating tumor-intrinsic role of CD276.

For my second concern, linking this biology to MERTK and efferocytosis, authors showed CD276 decreases phagocytic markers (lysosomal) which still only indirectly links the paper to TAM receptors. The effects on efferocytosis (lysosomal markers) provides better validation, and authors have changed language in the abstract that better suits that data presented. Overall, strengths outweigh weaknesses, and the overall paper is meritorious to this research filed.

Reviewer #2 (Remarks to the Author):

Unfortunately, the revised version of manuscript still needs some correction. Thus, the manuscript is missing updated Figures and Figure Legends. Authors should provide fully revised manuscript with included updated Figures and Supplemental materials. Otherwise, it is difficult for reviewers to comprehend the full revision in detail.

Comment 1: Authors still need properly respond to original questions:

Author's response to my 1st question: "Thank you for pointing out the discrepancy in the antibody treatment schedule in Fig. 7a and the missing specific information in the figure legend. We have corrected this in the updated manuscript."

I still have the same question. Where is updated Fig.7 and its Legend? I can't see any corrected legend.

2. Author's response to my 2nd question: We... conducted therapeutic experiments using MBT2 cells for C3/H mice and MB49 cells for C57/BL6 mice, which further confirmed the potential of CD276 in bladder cancer therapy. Additionally, we have included survival curve data, which provides essential information on the therapeutic efficacy of the proposed treatment (New Supplementary Fig. 7b, c").

Comment 2: Legends to New Supplementary Figures 7b,c are missing the experimental details: are these subcutaneous or orthotopic tumors? When treatment started: before or after tumor injection? What the protocol/schedule for treatment? What are dosages of administered antibodies?

Reviewer #3 (Remarks to the Author):

The authors have made efforts to answer the questions raised during the review process and it is very much appreciated. However, the data provided in the revised manuscript does not address some of the important concerns raised and therefore, does not give full confidence in the claims made in the manuscript.

1. The authors have provided analysis of TCGA analysis in response to the reviewers, however the data with BLCA looks most promising. To confirm that reported phenomenon is in fact a pan-cancer phenomenon, another relevant mouse model would be key. Otherwise, relevance of the findings to a particular type of cancer should be made clear in the title.

2. The authors made efforts to show the effect of CD276 KO specifically on macrophage infiltration however, the data presented does not confirm the KO is specific to TAMs only and thus does not

necessarily rule out the secondary effect of CD276 KO in other tumor-associated myeloid/immune cells on TAM infiltration. These aspects should be considered carefully, and authors should probably change the wording accordingly, to get more clarity.

3. In new fig. 4, is the expression is analyzed on the entire bladder. Since some of these proteins are also expressed by other infiltrating immune cells, it would be important to analyze the specific expression of the chemokine receptors specifically on TAM (possibly by flow cytometry) to gain direct mechanistic insight for reduced TAM infiltration

We would like to extend our heartfelt gratitude to the reviewers for their invaluable comments and insightful suggestions. We have taken their critiques and recommendations to heart and diligently addressed their concerns. In response to their feedback, we have added some new data and corrected our wording in the manuscript, effectively addressing the issues raised during the review period. The manuscript has benefited greatly from these efforts, resulting in a more robust and clear presentation of our novel findings. We would like to express our sincere appreciation to the reviewers for their critical role in improving the quality of our research. Additionally, we appreciate the reviewers' patience as the revision process required more time than anticipated due to the extensive experimental work involved. Once again, thank you to the reviewers for their valuable contributions, which have undoubtedly strengthened our research and improved the overall quality of our work.

REVIEWER COMMENTS

Reviewer #1 (Remarks to the Author):

This is a revised MS investigating role of B7 superfamily receptor CD276 in IO. To their credit, authors have successfully completely an important control investigating tumor-intrinsic role of CD276.

For my second concern, linking this biology to Mertk and efferocytosis, authors showed CD276 decreases phagocytic markers (lysosomal) which still only indirectly links the paper to TAM receptors. The effects on efferocytosis (lysosomal markers) provides better validation, and authors have changed language in the abstract that better suits that data presented. Overall, strengths outweigh weaknesses, and the overall paper is meritorious to this research filed.

Response: Thank you for your valuable feedback and constructive comments on our manuscript. We appreciate your careful review of our work and are pleased to hear that you find our research meritorious to the field. Regarding your second concern, we acknowledge the importance of providing a more direct link between our findings and TAM receptors and lysosome. Previous studies have shown that apoptotic cells activate the MerTK receptor-mediated signaling pathway in TAM, initiating TAM efferocytosis. In addition, studies have shown that lysosomal activation and fusion with phagosomes to form phagolysosomes to complete clearance of dead cells is one of the most critical steps in cytokinesis. Gerlach et al. found that clearance of apoptotic cells by macrophages (efferocytosis) promotes resolution signaling pathways that can be triggered by molecules derived from phagolysosomal degradation of apoptotic cells ¹. Wang et al. demonstrated that ensuring macrophage phagolysosomal maturation and redox homeostasis maintains their sustained enhanced efferocytosis ². Taken together, it is reasonable to assume that the lysosomal pathway is closely related to efferocytosis.

- 1 Gerlach, B. D. *et al.* Efferocytosis induces macrophage proliferation to help resolve tissue injury. *Cell Metab* **33**, 2445-2463 e2448, doi:10.1016/j.cmet.2021.10.015 (2021).
- 2 Wang, Y. T. *et al.* Metabolic adaptation supports enhanced macrophage efferocytosis in limited-oxygen environments. *Cell Metab* **35**, 316-331 e316, doi:10.1016/j.cmet.2022.12.005 (2023).

Reviewer #2 (Remarks to the Author):

Unfortunately, the revised version of manuscript still needs some correction. Thus, the manuscript is missing updated Figures and Figure Legends. Authors should provide fully revised manuscript with included updated Figures and Supplemental materials. Otherwise, it is difficult for reviewers to comprehend the full revision in detail.

Response: We sincerely appreciate your continued review of our manuscript. In response to your concern, we have promptly provided the fully revised manuscript with the updated Figures and Supplemental materials as per your suggestion to ensure clarity and comprehensibility.

Comment 1: Authors still need properly respond to original questions:

Author's response to my 1st question: "Thank you for pointing out the discrepancy in the antibody treatment schedule in Fig. 7a and the missing specific information in the figure legend. We have corrected this in the updated manuscript."

I still have the same question. Where is updated Fig.7 and its Legend? I can't see any corrected legend

Response: We sincerely apologize for the confusion regarding the corrected Figure 7 and its legend. Unfortunately, there seems to have been an oversight in the submission process, and the updated Figure 7 and its legend were not included. We uploaded two versions of the revised manuscript (with and without figures and legends), which may have caused the versioning error due to system reasons. We have re-uploaded the revised manuscript.

Figure 7. a. The experimental design of the bladder cancer tumorigenesis model and different strategies of treatment. The mice were treated with either anti-CD276 antibody (10mg/kg body weight, 3 times a week) with or without PD-1 antibody (200 μ g/mouse, day1, 3, 5, 7, 14) as indicated starting from week 22.

2. Author's response to my 2nd question: We... conducted therapeutic experiments using MBT2 cells for C3/H mice and MB49 cells for C57/Bl6 mice, which further confirmed the potential of CD276 in bladder cancer therapy. Additionally, we have included survival curve data, which provides essential information on the therapeutic efficacy of the proposed treatment (New Supplementary Fig. 7b, c").

Comment 2: Legends to New Supplementary Figures 7b,c are missing the experimental details: are these subcutaneous or orthotopic tumors? When treatment started: before or after tumor injection? What the protocol/schedule for treatment? What are dosages of administered antibodies?

Response: Thank you for your valuable comments. Regarding your concern about the lack of experimental details in the legends for the new Supplementary Figures 7b and 7c, we genuinely understand your point. To provide a clearer and more reproducible account of these treatment experiments, we are more than willing to offer additional experimental details. Subcutaneous MBT2/C3H or MB49/C57 bladder cancer model was established as described in New Supplementary Fig. 7a. At day 3 after injection, the mice with similar tumor volumes were randomly divided into four groups (six mice per group) for i.p. treatment with IgG (200 μ g/mouse at day 1, 3, 5, 7 and 14), anti-PD1 (200 μ g/mouse at day 1, 3, 5, 7 and 14), anti-CD276 (10mg/kg body weight; three times a week for 4 weeks) or anti-PD1 plus anti-CD276.

New Supplementary Fig. 7a Subcutaneous MBT2/C3H or MB49/C57 bladder cancer model. At day 3 after injection, the mice with similar tumor volumes were randomly divided into four groups (six mice per group) for i.p. treatment with IgG (200 μ g/mouse at day 1, 3, 5, 7 and 14), anti-PD1 (200 μ g/mouse at day 1, 3, 5, 7 and 14), anti-CD276 (10mg/kg body weight; three times a week for 4 weeks) or anti-PD1 plus anti-CD276.

Reviewer #3 (Remarks to the Author):

The authors have made efforts to answer the questions raised during the review process and it is very much appreciated. However, the data provided in the revised manuscript does not address some of the

important concerns raised and therefore, does not give full confidence in the claims made in the manuscript.

Response: We sincerely appreciate your diligence and valuable feedback during the review process. We are grateful for your recognition of our efforts to address the concerns raised in your initial review. Your feedback has been instrumental in improving the quality of our manuscript.

1. The authors have provided analysis of TCGA analysis in response to the reviewers, however the data with BLCA looks most promising. To confirm that reported phenomenon is in fact a pan-cancer phenomenon, another relevant mouse model would be key. Otherwise, relevance of the findings to a particular type of cancer should be made clear in the title.

Response: Thank you for your valuable comments. In response to your concern, we have revised the manuscript title to better reflect the focus of our study. The revised title is 'CD276 expression by tumor-associated macrophages promotes efferocytosis to confer immune evasion in bladder cancer'.

2. The authors made efforts to show the effect of CD276 KO specifically on macrophage infiltration however, the data presented does not confirm the KO is specific to TAMs only and thus does not necessarily rule out the secondary effect of CD276 KO in other tumor-associated myeloid/immune cells on TAM infiltration. These aspects should be considered carefully, and authors should probably change the wording accordingly, to get more clarity.

Response: Thank you for your comment. We agree that it is essential to clarify this aspect to provide a more accurate interpretation of our findings. To make the description clearer and more precise, we will revise the manuscript to explicitly state that our findings pertain primarily to TAMs while acknowledging the potential for secondary effects on other immune cells. We will also ensure that the wording accurately reflects this perspective throughout the manuscript.

3. In new fig. 4, is the expression is analyzed on the entire bladder. Since some of these proteins are also expressed by other infiltrating immune cells, it would be important to analyze the specific expression of the chemokine receptors specifically on TAM (possibly by flow cytometry) to gain direct mechanistic insight for reduced TAM infiltration.

Response: Thank you for your feedback. In the new Figures 4g and h, the samples utilized for Western blot analysis were specifically derived from flow-sorted tumor-associated macrophages (TAMs), rather than representing the entire bladder tissue. We have ensured that this important detail is accurately reflected in the revised manuscript.

REVIEWER COMMENTS

Reviewer #2 (Remarks to the Author):

The revised version of manuscript is significantly improved.

Reviewer #3 (Remarks to the Author):

The authors have made changes in the manuscript in response to question 1.

However the answers to question 2 and 3 still remain unanswered.

Regarding question 2: The authors claim that necessary changes will be made. However, I have been unable to spot the claimed changes. The authors should provide the clear link to the manuscript (page no, line no), where the necessary changes are made.

Regarding question 3: In response questions 3, the authors claim that "Western blot analysis were specifically derived from flow-sorted tumor-associated macrophages (TAMs), rather than representing the entire bladder tissue. We have ensured that this important detail is accurately reflected in the revised manuscript".

However, the methods section is revised manuscript states that "Mouse bladder tissues were removed and promptly lysed with ice-cold RIPA lysis buffer (P0013B, Beyotime, Jiangsu, China) with protease and phosphatase inhibitors...."

Therefore, it is not yet clear to me how western blot analysis was performed.

We sincerely thank the reviewers for their valuable comments and insightful suggestions. We have taken their criticisms and suggestions seriously and addressed their concerns diligently. The manuscript has benefited greatly from these efforts, making our new findings stronger and clearer. Once again, we thank the reviewers for their invaluable contributions, which have undoubtedly strengthened our research and improved the overall quality of our work.

Reviewer #2 (Remarks to the Author):

The revised version of manuscript is significantly improved.

Response: We sincerely thank the reviewers the helpful comments.

Reviewer #3 (Remarks to the Author):

The authors have made changes in the manuscript in response to question 1.

Response: We sincerely thank the reviewers the helpful comments.

However the answers to question 2 and 3 still remain unanswered.

Regarding question 2: The authors claim that necessary changes will be made. However, I have been unable to spot the claimed changes. The authors should provide the clear link to the manuscript (page no, line no), where the necessary changes are made.

Response: We sincerely appreciate your continued review of our manuscript. Perhaps due to versioning issues, the modified parts did not appear in the revised manuscript. To avoid ambiguity caused by versioning issues, we have made the following changes in the revised manuscript (revised section marked in red):

1. On page 8, lines 20-23 (lines 215-218), change to "We crossed them with *LysM-Cre*; *Rosa-tdTomato* mice to obtain *LysM-Cre*; *Rosa-tdTomato*; *CD276^{fl/fl}* homozygous mutant (CD276 cKO) mice with myeloid-specific CD276 deletion".
2. On page 8, lines 29-30 (lines 224-225), change to "The next step of our study was to investigate the effect of myeloid-specific CD276 on growth of spontaneous BLCA".
3. On page 24, lines 8-10 (lines 382-384), change to "We then crossed *ROSA-CD276* mice with *LysM-Cre* mice to obtain *LysM-Cre*; *ROSA-CD276* (CD276 cKI) mice with myeloid-restricted CD276 overexpression".

Regarding question 3: In response questions 3, the authors claim that "Western blot analysis were specifically derived from flow-sorted tumor-associated macrophages (TAMs), rather than representing the entire bladder tissue. We have ensured that this important detail is accurately reflected in the revised manuscript".

However, the methods section is revised manuscript states that "Mouse bladder tissues were removed and promptly lysed with ice-cold RIPA lysis buffer (P0013B, Beyotime, Jiangsu, China) with protease and phosphatase inhibitors...."

Therefore, it is not yet clear to me how western blot analysis was performed.

Response: Thank you for your careful reading and feedback on our manuscript. We are very sorry that due to our negligence we did not make timely corrections to the Methods section. Based on your suggestions, we have made the following changes to the Methods section of the manuscript (lines 709-711):

"The mouse bladder cancer tissues were dissociated into single cells, and TAMs were sorted by flow cytometry (see "Flow Cytometry Analysis and Fluorescence-Activated Cell Sorting" in the Methods section). TAMs were immediately lysed with ice-cold RIPA lysis buffer (P0013B, Beyotime, Jiangsu, China) with protease and phosphatase inhibitors (4693132001, Roche, Shanghai, China) using a gentleMACS Dissociator in order to identify protein expression (130-093-235, Macs Miltenyi Biotec, Guangzhou, China)."